# SCALING DIFFUSION LANGUAGE MODELS VIA ADAPTATION FROM AUTOREGRESSIVE MODELS

**Shansan Gong**[*1*], **Shivam Agarwal**[*2], **Yizhe Zhang**[3], **Jiacheng Ye**[1], **Lin Zheng**[1]
**Mukai Li**[1], **Chenxin An**[1], **Peilin Zhao**[4], **Wei Bi**[4], **Hao Peng**[2], **Jiawei Han**[2], **Lingpeng Kong**[1]
[1]The University of Hong Kong [2] University of Illinois at Urbana-Champaign
[3] Apple [4] Tencent AI Lab
sansa933@connect.hku.hk,shivama2@illinois.edu

## ABSTRACT

Diffusion Language Models (DLMs) have emerged as a promising new paradigm for text generative modeling, potentially addressing limitations of autoregressive (AR) models. However, current DLMs have been studied at a smaller scale compared to their AR counterparts and lack fair comparison on language modeling benchmarks. Additionally, training diffusion models from scratch at scale remains challenging. Given the prevalence of open-source AR language models, we propose adapting these models to build text diffusion models. We demonstrate connections between AR and diffusion modeling objectives and introduce a simple continual pre-training approach for training diffusion models. Through systematic evaluation on language modeling, reasoning, and commonsense benchmarks, we show that we can convert AR models ranging from 127M to 7B parameters (GPT2 and LLaMA) into diffusion models DiffuGPT and DiffuLLaMA, using less than 200B tokens for training. Our experimental results reveal that these models outperform earlier DLMs and are competitive with their AR counterparts. We release a suite of DLMs (127M-355M-7B) capable of generating fluent text, performing in-context learning, filling in the middle without prompt re-ordering, and following instructions. https://github.com/HKUNLP/DiffuLLaMA

## 1 INTRODUCTION

Large language models (LLMs) have ushered in a new era of artificial intelligence, demonstrating remarkable capabilities in generating high-quality text, in-context learning, and following complex instructions (OpenAI, 2023; Touvron et al., 2023a). These advancements are primarily rooted in the scaling up of autoregressive (AR) language models. During both training and inference, these models leverage vast datasets and billions of parameters, employing a strict left-to-right sequential process for memorization and generation. This approach has resulted in the emergence of intelligence capable of tackling diverse tasks (Wei et al., 2022a; Hoffmann et al., 2024). However, the ultimate upper limit of intelligence achievable through this paradigm remains an open question. While AR mechanisms form the foundation of current LLMs, they are not without limitations (Lin et al., 2021). Notable challenges include difficulties in future planning (Bachmann & Nagarajan, 2024; Hu* et al., 2024; Xie et al., 2024) and self-correction (Huang et al., 2024). These constraints have spurred researchers to explore alternative architectures for next-generation LLMs.

A compelling direction in current research focuses on the development of text diffusion models (Li et al., 2023b). Building upon the rapid evolution of diffusion models in various domains (Ho et al., 2020; Nichol & Dhariwal, 2021; Ramesh et al., 2021), innovative text diffusion models (Li et al., 2022; Lou et al., 2024) have opened up new possibilities for text generation. A unifying insight across these models is the potential of diffusion language models (DLMs) for controllable (Venkatraman et al., 2024), any-order, and parallel text generation (Gong et al., 2023a). Notably, DLMs exhibit promising capabilities in intermediate token correction (Ye et al., 2024b) and global planning (Zhang et al., 2023), thereby addressing key limitations inherent in the AR approach.

---

[*]Equal contribution

Despite the promising potential of text diffusion models, the relatively small model size limits the competitiveness of DLMs compared to AR models. Existing state-of-the-art DLMs such as Plaid 1B (Gulrajani & Hashimoto, 2023) and SEDD (Lou et al., 2024) are relatively small in size (127M-1B parameters) and under-trained, with less than 400B tokens of training data. This substantial gap in scale prevents fair comparisons with larger AR language models on many advanced capabilities and tasks, such as chain-of-thought reasoning abilities on complex mathematical benchmarks. Recent approaches (Ye et al., 2023) attempt adapt LLaMA models to DLMs based on masked language modeling (He et al., 2023). However, they find that the base model capabilities are lost during their adaptation stage. Pre-training at such a scale is extremely resource-intensive, and the challenge is even more pronounced for diffusion models. These models lack the computational optimizations that have been developed for LLMs (Samragh et al., 2024) and require significantly more resources than their AR counterparts, as noted by Gulrajani & Hashimoto (2023).

Given these scaling challenges, pre-trained LLMs emerge as an invaluable resource that we can leverage, considering the extensive computational efforts already invested in their development. This strategy aligns with recent trends where new models are scaled up or adapted to new architectures using existing LLMs (Wang et al., 2024; Zhang et al., 2024c). However, building DLMs through adaptation from AR models is non-trivial due to fundamental differences in their language modeling objectives. Two key distinctions present significant hurdles. First, AR models employ causal masking to prevent future information leakage, whereas diffusion models utilize bi-directional attention masks. Second, an AR LM processes clean inputs to predict subsequent tokens at each step, while a diffusion model operates on noisy inputs to predict their denoised versions.

To overcome these challenges, we propose a simple adaptation approach that bridges these discrepancies. We unify their modeling objectives (§3.2) and address the architectural differences by breaking the causal masking bias in AR models through attention mask annealing (§3.3). Additionally, we inherit the shift operation from AR models (§3.3). This streamlined adaptation recipe enables us to construct a pre-trained DLM that can effectively compete in the arena of LLMs. Building on this approach, we leverage the FineWeb (Penedo et al., 2024) and SlimPajama (Soboleva et al., 2023) pre-training corpora to continue training small and medium-sized DLMs based on GPT2 (Brown et al., 2020), and further train up to a 7B model based on LLaMA2 (Touvron et al., 2023b).

Our experiments provide a comprehensive comparison between AR LMs and DLMs across language modeling, reasoning, and infilling tasks. The evaluation encompasses diverse settings, including zero-shot, few-shot, and fine-tuning scenarios, addressing the limitations of relying solely on perplexity in previous works (Shi et al., 2024). Our contributions and empirical findings include:

- We demonstrate that by narrowing the gap between AR models and DLMs, it is possible to convert 127M-7B AR models (GPT2 and LLaMA2) into DiffuGPT and DiffuLLaMA with training on less than 200B tokens. Notably, DiffuGPT outperforms GPT2 in most tasks.

- We adapt 7B AR models to DLMs, greatly expanding the expertise compared to smaller-sized diffusion models. DiffuLLaMA emerges as the state-of-the-art DLM, exhibiting in-context learning, code generation, and strong infilling capabilities. Its generation speed is competitive with AR counterparts for unconditionally generating 1024 tokens using 256 diffusion timesteps.

- We provide a comprehensive benchmark for DLMs and release our adapted diffusion models (127M, 355M and 7B) along with open-source adaptation code, efficient fine-tuning scripts, and evaluation toolkits.

## 2 PRELIMINARY AND NOTATION

Diffusion models (Sohl-Dickstein et al., 2015; Song & Ermon, 2019; Ho et al., 2020; Song et al., 2021b) are latent variable generative models characterized by a forward and a reverse Markov process. We denote $\mathbf{x}_0 \sim p_{data}(\mathbf{x}_0)$ as the variable following the data distribution, and $\mathbf{x}_t \sim q(\mathbf{x}_t)$ as the noisy variable of $\mathbf{x}_0$ at time $t$, where the maximum time is $T$. The forward process $q(\mathbf{x}_{1:T}|\mathbf{x}_0) = \prod_{t=1}^{T} q(\mathbf{x}_t|\mathbf{x}_{t-1})$ corrupts the initial data $\mathbf{x}_0$ into a sequence of increasingly noisy variables $\mathbf{x}_{1:T}$. Accordingly, the backward Markov process models the joint probability as $p_\theta(\mathbf{x}_{0:T}) = p_\theta(\mathbf{x}_T) \prod_{t=1}^{T} p_\theta(\mathbf{x}_{t-1}|\mathbf{x}_t)$, which gradually denoises $\mathbf{x}_t$ to reconstruct the original data $\mathbf{x}_0$. Parameters $\theta$ are learned by minimizing the negative log-likelihood of $\mathbf{x}_0$, which can be opti-

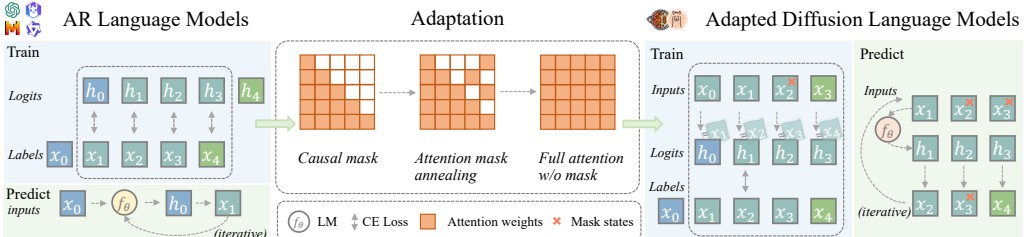

Figure 1: The overview of our approach to adapt autoregressive (AR) models to diffusion models. **Left**: The shift operation in AR models enables the output layer $h_i$ to approximate the distribution of next tokens $x_{i+1}$ in hidden representations through the cross entropy (CE) loss. **Middle**: We remove the causal mask gradually during training eventually making our model bi-directional. **Right**: inside the diffusion models we shift the logits to compute the loss with the next token (i.e., the loss on $h_i$ would be with respect to $x_{i+1}$), while perceptually, the diffusion models are still functioning as recovering the original signals (since $h_i$ corresponds to $x_{i+1}$ in AR loss).

mized through the evidence lower bound (ELBO),

$$-\log p_\theta(\mathbf{x}_0) \le \mathbb{E}_{q(\mathbf{x}_1|\mathbf{x}_0)}[-\log p_\theta(\mathbf{x}_0|\mathbf{x}_1)] + D_{\mathrm{KL}}(q(\mathbf{x}_T|\mathbf{x}_0)||p_\theta(\mathbf{x}_T)) + \mathcal{L}_T, \tag{1}$$

with $\mathcal{L}_T = \sum_{t=2}^T \mathbb{E}_{q(\mathbf{x}_t|\mathbf{x}_0)}[D_{\mathrm{KL}}(q(\mathbf{x}_{t-1}|\mathbf{x}_t,\mathbf{x}_0)||p_\theta(\mathbf{x}_{t-1}|\mathbf{x}_t))]$. For continuous text diffusion (Li et al., 2022; Gong et al., 2023b), at each forward step, perturbations are applied according to $q(\mathbf{x}_t|\mathbf{x}_{t-1}) = \mathcal{N}(\mathbf{x}_t; \sqrt{1-\beta_t}\mathbf{x}_{t-1}, \beta_t\mathbf{I})$, where $\beta_t \in (0,1)$ represents different scales across time steps such that $\mathbf{x}_T \sim \mathcal{N}(0, \mathbf{I})$. In the case of discrete denoising models (Ho et al., 2020; Austin et al., 2021; Zheng et al., 2024a), the forward process is defined as a categorical distribution $q(\mathbf{x}_t|\mathbf{x}_{t-1}) = \mathrm{Cat}(\boldsymbol{x}_t; \boldsymbol{Q}_t^\top \boldsymbol{x}_{t-1})$, where each $\mathbf{x}_t \in \{0,1\}^K$ is a one-hot vector with vocabulary size $K$, $\boldsymbol{Q}_t \in [0,1]^{K \times K}$ is the transition matrix, and each entry $[\boldsymbol{Q}_t]_{ij}$ denotes the probability of transition from the state $i$ to $j$. We build on the formulation of *absorbing discrete diffusion* (Austin et al., 2021), which specifies $\boldsymbol{Q}_t = (1-\beta_t)I + \beta_t \mathbf{1}\boldsymbol{m}^\top$. We denote $\mathbf{1}$ as an all-one vector of size $K$ and $\boldsymbol{m}$ as the one-hot encoding of a special [MASK] token in the vocabulary. Therefore, the transition matrix, $\boldsymbol{Q}_t$ indicates that with probability $1 - \beta_t$, $\boldsymbol{x}_t$ remains unchanged; otherwise, it transitions to $\boldsymbol{m}$, becoming absorbed into [MASK]. Letting $\overline{\boldsymbol{Q}}_t := \prod_{i=1}^t \boldsymbol{Q}_i = \alpha_t I + (1-\alpha_t)\mathbf{1}\boldsymbol{m}^\top$ and $\alpha_t := \prod_{i=1}^t (1-\beta_i)$, the distribution of $\boldsymbol{x}_t$ conditional on $\boldsymbol{x}_0$ is given by

$$q(\boldsymbol{x}_t|\boldsymbol{x}_0) = \mathrm{Cat}(\boldsymbol{x}_t; \overline{\boldsymbol{Q}}_t^\top \boldsymbol{x}_0) = \alpha_t I \boldsymbol{x}_0 + (1-\alpha_t)\boldsymbol{m}\mathbf{1}^\top \boldsymbol{x}_0 = \alpha_t \boldsymbol{x}_0 + (1-\alpha_t)\boldsymbol{m}, \tag{2}$$

since $\boldsymbol{x}_0$ is a one-hot vector and thus $\mathbf{1}^\top \boldsymbol{x}_0 = 1$. We expect $\alpha_T$ to approach $0$ such that the full noise data $\boldsymbol{x}_T$ equals $\boldsymbol{m}$ with probability 1.

The discrete time representation of $t \in [0,T]$, restricts $\boldsymbol{x}_t$ to fixed noise ratios. To avoid this bias and enable sampling from any noisy representation, we use continuous-time sampling, allowing $t$ to span any point within $[0,1]$ (Kingma et al., 2021; Shi et al., 2024; Zhao et al., 2024; Ou et al., 2024). Continuous-time sampling is equivalent to dividing $[0,1]$ into $T$ intervals and where $T \to \infty$. For any $0 \le s < t \le 1$, the forward process generalizes to $q(\mathbf{x}_t|\mathbf{x}_s)$. We will use this continuous-time notation in the following sections.

## 3 MODEL

We begin by formulating the continuous-time discrete diffusion process (§3.1) and establishing a connection between the discrete diffusion and autoregressive objectives (§3.2). Based on this equivalence, we propose an adaptation approach (§3.3) and a sampling algorithm (§3.4) for diffusion models adapted from AR models. The whole process is illustrated in Figure 1.

### 3.1 CONTINUOUS-TIME DISCRETE DIFFUSION PROCESSES

Following Eq.2 and $q(\boldsymbol{x}_t|\boldsymbol{x}_0) = \sum_{\boldsymbol{x}_s} q(\boldsymbol{x}_t|\boldsymbol{x}_s)q(\boldsymbol{x}_s|\boldsymbol{x}_0)$, the forward transition distribution between arbitrary points $s < t$ can be derived as

$$q(\boldsymbol{x}_t|\boldsymbol{x}_s) = \mathrm{Cat}(\boldsymbol{x}_t; \overline{\boldsymbol{Q}}_{s|t}^\top \boldsymbol{x}_s) = \frac{\alpha_t}{\alpha_s}\boldsymbol{x}_s + (1 - \frac{\alpha_t}{\alpha_s})\boldsymbol{m}, \tag{3}$$

with $\overline{\boldsymbol{Q}}_{s|t} := \overline{\boldsymbol{Q}}_s^{-1}\overline{\boldsymbol{Q}}_t = \frac{\alpha_t}{\alpha_s}I + (1 - \frac{\alpha_t}{\alpha_s})\mathbf{1}\boldsymbol{m}^\top$. The corresponding backward transition distribution conditional on $\boldsymbol{x}_0$ is also available in closed form,

$$q(\boldsymbol{x}_s|\boldsymbol{x}_t, \boldsymbol{x}_0) = \frac{q(\boldsymbol{x}_t|\boldsymbol{x}_s)q(\boldsymbol{x}_s|\boldsymbol{x}_0)}{q(\boldsymbol{x}_t|\boldsymbol{x}_0)} = \begin{cases} \frac{\alpha_s - \alpha_t}{1 - \alpha_t}\boldsymbol{x}_0 + \frac{1 - \alpha_s}{1 - \alpha_t}\boldsymbol{m} & \text{if } \boldsymbol{x}_t = \boldsymbol{m}, \\ \boldsymbol{x}_0 & \text{if } \boldsymbol{x}_t \neq \boldsymbol{m}. \end{cases} \qquad (4)$$

In discrete diffusion processes, we aim to approximate the backward transition distribution $q(\boldsymbol{x}_s|\boldsymbol{x}_t, \boldsymbol{x}_0)$ using a denoising model $p_\theta(\boldsymbol{x}_s|\boldsymbol{x}_t, f_\theta(\boldsymbol{x}_t))$, where $f_\theta(\boldsymbol{x}_t)$, an approximation of $\boldsymbol{x}_0$, is usually the output of neural networks such as a transformer (Vaswani et al., 2017). We can define the denoising model to have a similar form of backward transitions as $p_\theta(\boldsymbol{x}_s|\boldsymbol{x}_t) = \frac{\alpha_s - \alpha_t}{1 - \alpha_t}f_\theta(\boldsymbol{x}_t) + \frac{1 - \alpha_s}{1 - \alpha_t}\boldsymbol{m}$. According to the training objective in Eq.1, the KL-divergence of $\mathcal{L}_T$ at each step $t$ can be simplified to a reweighted cross-entropy function,

$$D_{\mathrm{KL}}(q(\boldsymbol{x}_s|\boldsymbol{x}_t, \boldsymbol{x}_0)||p_\theta(\boldsymbol{x}_s||\boldsymbol{x}_t)) = -\frac{\alpha_s - \alpha_t}{1 - \alpha_t}\delta_{\boldsymbol{x}_t, \boldsymbol{m}}\boldsymbol{x}_0^\top \log f_\theta(\boldsymbol{x}_t), \qquad (5)$$

where $\delta_{a,b}$ is the indicator function for $a = b$. If we take the limit and let $T \to \infty$, the first two terms of Eq.1 will approach 0 and some constant, respectively. Thus the evidence lower bound (ELBO) effectively becomes $\mathcal{L}_T$ and

$$\lim_{T \to \infty} \mathcal{L}_T = \int_0^1 \frac{\alpha_t'}{1 - \alpha_t}\mathbb{E}_{q(\mathbf{x}_t|\mathbf{x}_0)}[\delta_{\boldsymbol{x}_t, \boldsymbol{m}}\boldsymbol{x}_0^\top \log f_\theta(\boldsymbol{x}_t)]\, dt. \qquad (6)$$

The full derivation is listed in Appendix A.2. The same form of ELBO which is invariant to noise schedule but related to the signal-to-noise ratio (SNR) is also introduced in Kingma et al. (2021); Shi et al. (2024). Following Austin et al. (2021), we choose the noise schedule $\alpha_t = 1 - t$, then $\frac{-\alpha_t'}{1 - \alpha_t} = \frac{1}{t}$. The previous discussion focused on the single token $\boldsymbol{x}_t$, and can be applied independently to a text sequence of $N$ tokens $\mathbf{x}_t = [\boldsymbol{x}_t^1, \boldsymbol{x}_t^2 \dots, \boldsymbol{x}_t^N]$. During training, we do not compute integral loss in Eq.6 for efficiency consideration; instead, we sample $t$ for each data point. The final loss at $t$ is

$$\mathcal{L}_t^{1:N} = \frac{1}{t}\mathbb{E}_{q(\mathbf{x}_t|\mathbf{x}_0)}\left[-\sum_{n=1}^N \delta_{\mathbf{x}_t^n, \boldsymbol{m}}(\mathbf{x}_0^n)^\top \log f_\theta(\mathbf{x}_t^{1:N})_n\right], \qquad (7)$$

where $f_\theta(\mathbf{x}_t^{1:N})_n$ denotes the whole input sequence is fed into the transformer model and the $n$-th output token is indexed.

## 3.2 Unifying Language Modeling Objectives

The training objective of autoregressive (AR) language models is the negative log-likelihood of each ground-truth token provided the preceding tokens,

$$\mathcal{L}_{AR}^{1:N} = -\sum_{n=1}^N (\mathbf{x}_0^n)^\top \log f_\theta(\mathbf{x}_0^{1:n-1})_{n-1}. \qquad (8)$$

Comparing Eq.8 against Eq.7, we note that while both take the form of cross-entropy functions, Eq.7 includes an additional reweighting term $\frac{1}{t}$ and an indicator function $\delta_{\mathbf{x}_t^n, \boldsymbol{m}}$. They result from the definition of discrete diffusion processes (§3.1). The reweighting emphasizes smaller $t$ where $\mathbf{x}_t$ contains fewer masked tokens, and this can be regarded as the importance sampling (Nichol & Dhariwal, 2021). The indicator specifies which tokens are masked for prediction. The AR training objective Eq.8, on the other hand, constrains the context to be unidirectional via attention masking and shifts the targets so that each token predicts the next token instead of itself. These discrepancies form the basis of our adaptation framework, which is detailed in §3.3.

In fact, an alternative way to understand AR modeling, through the lens of diffusion models, is to consider a diffusion process where the forward pass deterministically masks right-to-left and token-by-token (Austin et al., 2021; Hoogeboom et al., 2022). This yields a backward process generating one token at a time from left to right, running with $T = N$ denoising steps in total. As discussed in Austin et al. (2021), the loss objective of this diffusion process is equivalent to standard cross-entropy (Eq.8) commonly used to train AR language models. This crafted diffusion process for AR models represents a special case of discrete diffusion (§3.1), yet it is limited to unidirectional context and sequential token generation. In contrast, general discrete diffusion processes can leverage bidirectional context and support parallel generation in arbitrary orders.

## 3.3 ADAPTATION

Building on the connection between AR modeling and discrete diffusion processes, we construct an adaptation recipe next. Figure 1 shows an overview of our adaptation approach. We use attention mask annealing, shift operations, and a time-embedding free architecture to narrow the differences between AR and DLMs.

---

**Algorithm 1** Adaptation Training

1: **Input:** network $f_\theta$ initialized by existing models, training corpus $p_{data}(\boldsymbol{x}_0^{1:N})$, mask token $\boldsymbol{m}$.
2: **Output:** model parameters $\theta$.
3: **repeat**
4:     Draw $\boldsymbol{x}_0^{1:N} \sim p_{data}$ and set *labels* $\leftarrow \boldsymbol{x}_0^{1:N}$
5:     Sample $t \in Uniform(0, 1)$
6:     Sample $\boldsymbol{x}_t^{1:N} \sim q(\boldsymbol{x}_t|\boldsymbol{x}_0)$
7:     Anneal the attention mask *attn_mask*
8:     Forward *logits* $\leftarrow f_\theta(\boldsymbol{x}_t^{1:N})$ with *attn_mask*
9:     Right shift *logits* by one position
10:    $\mathcal{L}_t = \frac{1}{t}\delta_{x_t,m}\text{CE}(logits, labels) \triangleright$ Eq.7
11:    Backprop with $\mathcal{L}_t$ and update $\theta$
12: **until** end training

**Algorithm 2** Sampling

1: **Input:** Trained diffusion model $f_\theta$, sampling algorithm $\tau$, mask token $\boldsymbol{m}$, start token $\boldsymbol{s}$.
2: **Output:** generated sample $\boldsymbol{x}_0$.
3: **Initialize** $\boldsymbol{x}_T^{1:N} = \boldsymbol{m}$.
4: **for** $t = T, \dots, 1$ **do**
5:     Forward *logits* $\leftarrow f_\theta(\boldsymbol{x}_t^{1:N})$
6:     Sample $\tilde{\boldsymbol{x}}_0^{1:N} \sim Categorical(\tau(logits))$
7:     **for** $n = 1, \dots, N$ **do**
8:        $\boldsymbol{x}_{t-1}^n = q(\boldsymbol{x}_{t-1}^n|\boldsymbol{x}_t^n, \tilde{\boldsymbol{x}}_0^n) \triangleright$ Eq.4
9:     **end for**
10:    Right shift $\boldsymbol{x}_{t-1}^{1:N} = [\boldsymbol{s}, \boldsymbol{x}_{t-1}^{1:N-1}]$
11: **end for**
12: **Return** $\boldsymbol{x}_0^{2:N}$

---

**Attention Mask Annealing** The prediction of the $n$-th token, given all preceding tokens, $f_\theta(\mathbf{x}_0^{1:n-1})$, is usually implemented by causal attention masking in transformer-based AR language models. As shown in Figure 1, causal attention masks set all entries in the upper triangle of the self-attention matrices to zero, so each token cannot attend to its respective future tokens. Such causal masking prevents the model from learning right-to-left dependencies for more general diffusion processes. To address this limitation while preserving left-to-right conditionals during adaptation, we introduce an incremental annealing process from causal masks to full attention matrices. During annealing, the causal mask is not immediately removed; instead, it is retained at a controlled ratio, as shown in the middle part of Figure 1. At each training step, we sample the amount of context from the right side and progressively increase this amount till we obtain the full attention mask.

**Shift Operation** AR models also apply a shifting operation, where the target output is the input sequence shifted left by one position. In other words, the prediction target of the $(n-1)$-th token is the $n$-th token, contrasting with typical diffusion models that try to predict masked tokens at their original positions. When initializing text diffusion models with AR model parameters, the model would tend to output the hidden representations of the shifted input sequence. If we continue to optimize the cross-entropy objective based on the original token positions, the model struggles to adapt due to misalignment between input and output. Instead, we maintain the shift operation (Algo.1, line 9), treating the output logits at each position as corresponding to the next token. When calculating the objective, we align prediction targets so that the diffusion model learns to recover the original signals. This process is illustrated in the right panel of Figure 1.

**Time-Embedding-Free Architecture** Many diffusion models for text generation (Li et al., 2022; Dieleman et al., 2022; Gulrajani & Hashimoto, 2023; Lou et al., 2024; Shi et al., 2024) incorporate time embedding layers to represent the information of current timesteps $t$, which can explicitly indicate the noise scale of the input noisy data. While inferring these timesteps can be challenging for image diffusion models (Ho et al., 2020; Li et al., 2024), some discrete text diffusion models (He et al., 2023) assert that timesteps $t$ can be easily learned implicitly based on the number of mask tokens. Since AR models are not equipped with time embedding layers, we also choose not to use the time embedding, resulting in no additional parameters compared to previous diffusion models.

## 3.4 SAMPLING

Following Shi et al. (2024), we initialize $\boldsymbol{x}_T$ with all `[MASK]` tokens and then sample tokens according to the time reversal $q(\boldsymbol{x}_s|\boldsymbol{x}_t, \boldsymbol{x}_0)$ in Eq.4. At each timestep, if $\boldsymbol{x}_t$ is a mask, it will jump to the predicted $\boldsymbol{x}_0$ at time $s$ with probability $\frac{\alpha_s - \alpha_t}{1 - \alpha_t}$. After $T$ iterations, the model generates the full sequence. Since our adapted models are trained with the shift operation, at each sampling iteration,

we shift back the generated sentence and prepend a start token before the next forward pass (Algo.2, line 10). Usually larger $T$ requires more interactions of computation, and can yield texts in higher quality, and this trade-off can be controlled easily through $T$. Through experiments, we find that the output generated by diffusion models is diverse and scattered. Therefore, for conditional generation tasks, we improve the sampling procedure to ensure that only tokens with high probabilities from neural networks are denoised (Ghazvininejad et al., 2019; Chang et al., 2022; Zheng et al., 2024a), so that the model could predict tokens mostly relevant to the input. In addition, existing sampling techniques for AR language models, including top-$k$ and nucleus sampling (Holtzman et al., 2020), can be seamlessly applied to diffusion models as well.

# 4 EXPERIMENT

## 4.1 ADAPTATION SETUP

**DiffuGPT** We use the 30 billion tokens[1] random split from the FineWeb dataset (Penedo et al., 2024), an improved corpus than OpenWebText (Gokaslan & Cohen, 2019) used in prior DLMs (Lou et al., 2024), to continue training GPT2 base (Radford et al., 2019). We use sequence packing, logits shifting, and 10K-step attention mask annealing to transform GPT2 to DiffuGPT.

**DiffuLLaMA** We continue pre-training LLAMA-2-7-HF (Tou-vron et al., 2023a) on a mixture of SlimPajama (70%) (Soboleva et al., 2023) and Starcoder (30%) (Li et al., 2023a) data following TinyLLaMA (Zhang et al., 2024a). We randomly sample 65 billion tokens from this mixture and use sequence packing with context length of 2048. For efficient implementation we enable flash-attention 2 (Dao, 2024) and directly use bi-directional attention without attention mask annealing.

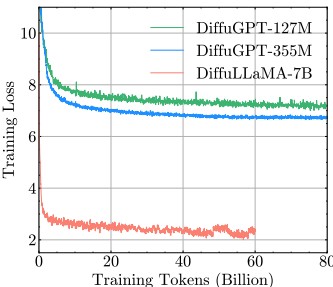

For both adaptation settings, we employ full parameter finetuning with `bf16`. Please refer to Appendix B.2 for details. We plot the training loss curve in Figure 2. We train DiffuLLaMA on 60B tokens and achieve a lower loss compared to 127M and 335M models, suggesting a scaling trend similar to that of AR LLMs (Kaplan et al., 2020). We also note that there is still scope for training more, since the model does not show signs of saturation.

Figure 2: Training loss over tokens for various model sizes of our adapted diffusion models.

## 4.2 EVALUATION SETUP

Previously developed diffusion language models (Gulrajani & Hashimoto, 2023; Lou et al., 2024; Shi et al., 2024; Ou et al., 2024) evaluate model performance using zero-shot perplexity on benchmark datasets. However, this metric alone does not fully capture a model's capabilities for several reasons. First, lower perplexity does not always correlate with human-like content, even in autoregressive models (Kuribayashi et al., 2021). Additionally, the loss from text diffusion models only indicates an upper bound on negative log-likelihood. While Kingma et al. (2021); Shi et al. (2024) demonstrate that the ELBO is invariant to the noise scheduler, discrepancies between continuous diffusion, discrete diffusion, and autoregressive loss still hinder fair comparisons across different model types. Given the ample evaluation benchmarks (Gu et al., 2024) for LLMs, we propose a more comprehensive evaluation for diffusion models.

**Tasks and Metrics** We consider TriviaQA (Joshi et al., 2017) to test the reading comprehension of models and last word completion task Lambada (Paperno et al., 2016) to test how models capture long-range dependencies in text. These two tasks are measured by exact match accuracy. We also test for common sense reasoning tasks HellaSwag (Zellers et al., 2019), Winogrande (Sak-aguchi et al., 2021), SIQA (Sap et al., 2019) and PIQA (Bisk et al., 2020), all of which involve multiple-choice questions assessed by accuracy. On grade school math problems GSM8K (Cobbe et al., 2021), we follow Ye et al. (2024b) in finetuning setting using the augmented symbolic data to test the CoT (Wei et al., 2022b) math reasoning abilities of diffusion models. Following Shen

---

[1]This is the total number of tokens used; however, our effective training tokens exceed this count, meaning that we train for more than one epoch.

Table 1: Comprehensive evaluation of different diffusion language models and the same size pre-trained autoregressive models. There are 3 types of these models: AR for autoregressive, DD for discrete diffusion and CD for continuous diffusion. For the infilling task, we use ROUGE-1/2/L score; for other tasks, we use the accuracy (%) metric. * indicates we finetune GSM8K on models; other tasks are all in zero-shot setting. Numbers in the () indicate that AR models are only given prefix for infilling tasks. We bold the best performance among diffusion language models and underline results that surpass their base models.

| Model | Size | Type | QA TriQA | Word Lamb. | CommonSense Reasoning | | | | Math GSM8K* | Infilling ROCStories | Code |
|---|---|---|---|---|---|---|---|---|---|---|---|
| | | | | | HSwag | Wino. | SIQA | PIQA | | | |
| GPT2-S | 127M | AR | 4.0 | 25.9 | 29.9 | 48.5 | 35.7 | 62.1 | 44.8 | (7.8/0.8/7.4) | (1.6) |
| SEDD-S | 170M | DD | 1.5 | 12.4 | 30.2 | 50.1 | 34.4 | 55.6 | 45.3 | 11.9/0.7/10.9 | 0.7 |
| DiffuGPT-S | 127M | DD | 2.0 | 45.0 | 33.4 | 50.8 | 37.0 | 57.7 | 50.2 | 13.7/1.4/12.6 | 0.3 |
| GPT2-M | 355M | AR | 6.7 | 37.7 | 38.3 | 50.7 | 37.7 | 67.4 | 45.6 | (8.6/0.9/8.2) | (2.6) |
| SEDD-M | 424M | DD | 1.8 | 23.1 | 31.5 | 49.0 | 35.4 | 56.1 | 53.5 | 13.1/1.4/12.2 | 0.5 |
| DiffuGPT-M | 355M | DD | 3.8 | 60.5 | 37.2 | 52.6 | 39.0 | 59.6 | 61.8 | 18.7/2.7/17.0 | 2.9 |
| Plaid1B | 1.3B | CD | 1.2 | 8.6 | 39.3 | 51.3 | 32.3 | 54.5 | 32.6 | 12.1/1.1/11.2 | 0.1 |
| LLaMA2 | 7B | AR | 45.4 | 68.8 | 74.9 | 67.1 | 44.8 | 78.3 | 58.6 | (11.6/2.1/10.5) | (1.7) |
| DiffuLLaMA | 7B | DD | 18.5 | 70.9 | 58.7 | 56.4 | 43.2 | 63.3 | 63.1 | 23.3/5.5/21.2 | 15.5 |

et al. (2023), we also test the story infilling tasks using ROCStories (Mostafazadeh et al., 2016) and evaluate using ROUGE score (Lin, 2004). To test the code infilling, we adopt Humaneval (Bavarian et al., 2022a) single line infilling task, which is evaluated by pass@1 rate. We evaluate DiffuLLaMA's math reasoning and in-context learning ability by evaluating on MAWPS (Koncel-Kedziorski et al., 2016) consisting of math word problems and SATMATH from AGI-eval consisting of math problems from SAT exam (Zhong et al., 2024). We base our implementation on `lm-evaluation-harness` (Gao et al., 2024) and re-implement all tasks across models to ensure a fair comparison.

**Implementation Details** For pre-trained diffusion language models, we mainly use continuous diffusion (CD) model Plaid 1B (Gulrajani & Hashimoto, 2023), discrete diffusion (DD) model SEDD (Lou et al., 2024) with different sizes as baselines. MD4 (Shi et al., 2024) and RADD (Ou et al., 2024) are based on and compared with SEDD, so we mainly compare SEDD. For autoregressive (AR) baselines, we consider the base models from which our models adapt. We implement infilling tasks for AR models by feeding the prefix and cutting off the generation length using the oracle length, considering that these AR models are not supporting infilling. For the sentence completion task, $T$ is the exact number of ground truth tokens for DD and 32 for CD. For 4 multi-choices tasks from commonsense reasoning, we compute the loss (Eq.6) of each choice (averaged by token) and choose the one with lowest loss (perplexity). For GSM8K finetuning, we use parameter-efficient LoRA tuning (Hu et al., 2022) for DiffuLLaMA. The decoding $T$ are set to 32 by default. The detailed settings are in Appendix B.3.

## 4.3 LANGUAGE MODELING CAPACITIES

**Benchmark performance** According to Table 1, the results on diverse tasks demonstrate that our adapted diffusion models achieve the state-of-the-art results among all existing diffusion language models (DLMs). We observe that diffusion models with larger parameters show improved performance, likely due to better base AR models. DiffuLLaMA's performance still falls short of the LLaMA2 model. This drop in performance is likely because DiffuLLaMa is trained on a small subset of SlimPajama and Starcoder data. We believe more training tokens can help improve these numbers. TriviaQA and PIQA are significant challenging for DLMs, probably because they require specific physical knowledge, such as *the capital of a city* or *the boiling point of water*; while our models are trained on 30B-70B tokens, which may be insufficient to preserve the general knowledge in the original LMs (Ke et al., 2023).

In tasks that require more extensive global reasoning, such as complex mathematics and coding, DLMs consistently exhibit better performance compared to AR models that rely solely on left-to-

right modeling capabilities. Remarkably, DLMs demonstrate their strengths in infilling tasks. Regular LLMs like LLaMA2 are not trained for filling-in-the-middle (FIM) tasks like those in Roziere et al. (2023), making them incapable of handling infilling. Considering this, we do not provide the suffix information to the model, which might result in an unfair comparison. But the FIM requires re-arranging the order of pre-training/inference sequence with special tokens (Zheng et al., 2024b), while diffusion training naturally supports this in its objective modeling.

Tasks in Table 1 mainly measure conditional modeling abilities, where Plaid 1B performs unsatisfactorily for conditional generation tasks even though with 1B parameters. We attribute this result to the gap between the continuous diffusion modeling and discrete text representation; in contrast, discrete diffusion models align more closely with AR modeling, naturally supporting conditional generation. Despite this, as illustrated in Figure 3, Plaid 1B demonstrates its strength in unconditional generation, highlighting its language modeling capabilities as a generative model. These findings reveal that the previous evaluation based on the perplexity of test data is too general to accurately assess the model's true capabilities, while our evaluation offers a more nuanced benchmark.

**Unconditional Generation**   We evaluate the quality of text unconditionally generated by DLMs in Figure 3. The perplexity is measured using GPT2 large, consistent with the prior work (Lou et al., 2024), where the data of MD4 (Shi et al., 2024) is sourced from its original paper. To make sure low perplexity is not brought by repeated content, we assess the distinct 2-gram diversity of the generated text. Our model achieves low perplexity while maintaining a high level of diversity, validating that the DiffuGPT series excels in fluent text generation. As the number of decoding steps increases, thereby extending the test computing time, the fluency of unconditional generation improves. Similarly, increasing model size also contribute to better performance. An increase in generation perplexity is often associated with a slight decrease in diversity, which is a common phenomenon. Notably, DiffuGPT outperforms both SEDD and MD4 mod-

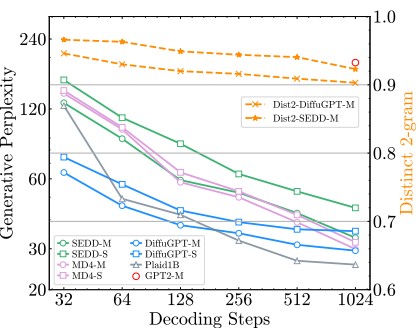

Figure 3: Quality evaluation for unconditional generation, with perplexity measured by GPT2 large and distinct 2-gram diversity.

els, particularly at lower step counts (e.g., 64 steps), while as the continuous diffusion models, Plaid 1B needs more decoding steps to generate more fluent texts. DiffuGPT thus exhibits a significant advantage on less sampling time. We outline the decoding hyperparameters and show the diversity changes across different settings in Appendix C.1, which also includes generation cases.

## 4.4   ANALYSIS ON DIFFULLAMA

We validate that increasing the size of adapted DLMs significantly enhances the performance of downstream tasks in Table 1. Further, we aim to assess if the 7B model demonstrates in-context learning and reasoning capabilities similar to AR LLMs. Table 2 presents the exact match accuracy between gold labels and predictions generated by DiffuLLaMA across zero-shot (ZS), few-shot (FS), and FS with chain-of-thought (CoT) scenarios. Besides, we deploy the self-consistency approach (Wang et al., 2023), considering that small DLMs can indeed benefit from this technique (Ye et al., 2024b). We use majority vote to choose the best answer from 3 individual predictions, and also report

Table 2: Performance on math/QA benchmarks (↑). We compare of DiffuLLaMA with zero-shot (ZS), few-shot (FS), self-consistency (SC), hit@$k$ and chain-of-thought (CoT) prompts.

| Settings | MAWPS | SATMath | TriviaQA |
|---|---|---|---|
| LLaMA2 | 63.5 | 24.5 | 45.4 |
| DiffuLLaMA-ZS | 9.7 | <1 | 18.5 |
| DiffuLLaMA-FS | 31.3 | 23.6 | 20.9 |
| DiffuLLaMA-SC | 33.1 | 27.7 | 26.0 |
| DiffuLLaMA-@$k$ | 40.8 | 57.7 | 34.1 |
| DiffuLLaMA-CoT | 28.7 | 9.5 | - |

the hit rate @$k$ with $k = 3$, which measures whether any of the $k$ predictions include the correct answer, serving as a reference for the model's upper bound. For in-context learning (ICL) evaluations, we give 4-shot on math tasks and 2-shot on TriviaQA.

The performance improvement from zero-shot to few-shot settings suggests that DiffuLLaMA can learn from ICL examples, particularly in following to the format of answers as we observe. We hypothesize that the adapted model retains some of the abilities from the base AR model. We randomly select the ICL demonstration here and anticipate that advanced ICL strategies in LLMs (Wu et al., 2023) could yield potentially higher results. The self-consistency offers LMs with an effective approach to test-time scaling (Snell et al., 2024), and DiffuLLaMA shows that it can also leverage this method. Furthermore, we report the hit rate results in generated candidate answers, highlighting the model's potential to produce the correct answer. This reveals that the current model exhibits high uncertainty about its responses, leading to temporarily suboptimal performance. We also observe that adding step-wise solutions in the in-context example (CoT) leads to a drop in performance, likely due to the absence of instruction tuning, similar to the findings in LLMs (Ouyang et al., 2022). We will leave instruction tuning as the future work as Ye et al. (2023) show that text diffusion model can benefit from instruction tuning. In summary, we show the potential capabilities of DiffuLLaMA, which motivates us to further investigate the scaling of diffusion models.

## 4.5 DISCUSSIONS

**Ablation Test on GSM8K-symbolic**  Direct ablation on adaptation training is costly; hence, we conduct preliminary experiments to determine the adaptation recipes. Following Ye et al. (2024b), we finetune models on the augmented GSM8K symbolic dataset using various base models and training objectives. The models are either trained from scratch (random initialization) or initialized with GPT2-S/M weights. Training objectives includes autoregressive training with a causal mask, continuous diffusion loss (CD), and discrete diffusion loss (DD). As shown in Table 3, different training objectives yield comparable results when training from scratch. However,

Table 3: Ablation test for adaptation approaches on GSM8K symbolic dataset. CD is for continuous diffusion and DD is for discrete diffusion.

| Base models | Random | GPT2-S | GPT2-M |
|---|---|---|---|
| Autoregressive | 30.5 | 44.8 | 45.6 |
| CD | 27.9 | 19.2 | 20.2 |
| DD-w/o shift | - | 33.5 | 34.5 |
| DD-w/o anneal | - | 43.3 | 47.2 |
| DD | 28.0 | 45.4 | 49.7 |

when using GPT2 as the base model, the CD loss performs worse than both the DD and AR losses. We attribute this to the better alignment of DD and AR losses as discussed in §3.2. Previous continuous diffusion models (Dieleman et al., 2022; Gulrajani & Hashimoto, 2023) has reparameterized the estimation of embeddings into the CE loss. However, adapting diffusion models from an AR model in continuous space necessitates an additional projection from the embedding to a categorical distribution, increasing the difficulty of adaptation.

For DD loss, removing attention mask annealing and shift operations both degrade performance, indicating the efficacy of our approaches. The mask annealing has minimal impact, so we choose to omit it for 7B adaptation to simplify implementation using flash-attention 2.

Direct DD loss finetuning on GPT2 achieves accuracy of 45.4 and 49.7 for small and medium models, respectively, outperforming GPT2 AR finetuning. However, finetuning from already adapted diffusion language models (DiffuGPT) yields accuracy of 50.2 and 61.8 (Table 1). This demonstrates the superiority of DiffuGPT as the current best diffusion base model at this size and highlights that a better base model leads to improved results. Even with the same DD loss, DiffuGPT's finetuning converges faster and achieves lower loss, as shown in Appendix C.2.

**Inference Speed**  AR models usually utilize key-value caching (incremental decoding; Ott et al. 2019) to enhance throughput during decoding. However, due to the nature of sequential token generation, they are highly memory-bound and cannot fully exploit modern accelerators (Chen et al., 2023). In contrast, diffusion models, despite not having a

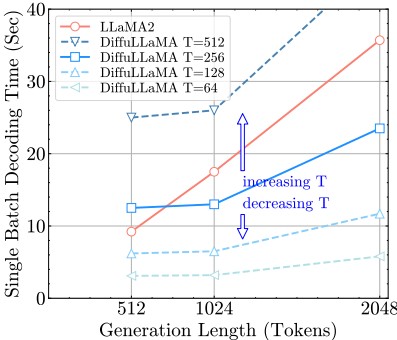

Figure 4: Single batch decoding speed (seconds) for different models using flash-attention 2.

concept of caching and requiring self-attention over the entire sequence at each iteration, can operate with fewer iterations than the sequence length and exhibit less memory-bound behavior. Their performance can be further boosted with hardware-aware optimizations like flash-attention (Dao et al.,

2022; Dao, 2024; Shah et al., 2024). In Figure 4, we evaluate the decoding latency with batch size 1 using flash-attention 2 and illustrate that our DiffuLLaMA achieves better inference efficiency using $T = 256$ when generating sequences of length 1024 or longer. This underscores the significant potential of diffusion models for efficient inference. Further decreasing $T$ can lead to faster decoding but may sacrifice quality. Additional latency comparisons are provided in Appendix C.5.

## 5 RELATED WORK

**Continue Pre-training** Continue pre-training is commonly used in adapting an existing language model (LM) to a domain-specific LM (Ke et al., 2023) or enabling new abilities of LM, such as for longer context (Chen et al., 2024) or code generation (Xu et al., 2024). Pre-training LMs is non-trivial and expensive (Samragh et al., 2024), thus in exploring of new architectures of LMs such as Mamba (Gu & Dao, 2023) and gated attention, Wang et al. (2024); Zhang et al. (2024c) choose to transfer from LMs to save the training cost. However, all these continue pre-training works follow the autoregressive (AR) language modeling, while adapting LMs into diffusion language model is more challenging due to discrepancies between their modeling objectives.

**Text Diffusion Models** Diffusion models have demonstrated significant diversity and controllability in image generation (Ho et al., 2020; Song et al., 2021a; Ramesh et al., 2022). Building on this success, line of research (Li et al., 2022; Gong et al., 2023b;a; Dieleman et al., 2022) build continuous diffusion models for text generation tasks. Among them, Lin et al. (2023) experiment with a pre-training and finetuning framework under a small scale; Gulrajani & Hashimoto (2023) highlight the scaling law of continuous diffusion models, revealing that the compute-optimal requires longer training than their AR counterparts. To address the discrete nature of text, Austin et al. (2021); Hoogeboom et al. (2021); Zheng et al. (2024a) incorporate an absorbing [MASK] state as noise, laying the foundation for discrete diffusion models, which are further developed by Lou et al. (2024); Shi et al. (2024); Ou et al. (2024); Zhao et al. (2024); Sahoo et al. (2024). By connecting text diffusion models with pre-trained masked language models (MLMs; Devlin et al. 2019), Ye et al. (2023); He et al. (2023) initialize discrete diffusion models using MLMs. Besides, the unification between diffusion and AR generation is also discussed in image generation (Li et al., 2024). However, the adaptation of diffusion models from AR LLMs remains unexplored.

**Non-autoregressive Generation** Non-autoregressive (NAR) models, introduced by Gu et al. (2018), break free from the left-to-right generation constraint, allowing for new capabilities like planning with future tokens (Wu et al., 2024). Current diffusion language models are a notable part of the NAR family (Gong et al., 2023b). Given the challenges of developing NAR models, researchers often seek to find a trade-off. For instance, SSD-LM (Han et al., 2023) leverages diffusion models to iteratively generate text blocks, facilitating a semi-NAR generation process. Similarly, CLLM (Kou et al., 2024) enhances LLMs by enabling the parallel generation of $n$ tokens, thereby improving decoding speed. FiLM (Shen et al., 2023) adapts language models to generate tokens in any order, which is particularly useful for infilling tasks. Guo et al. (2020) trains a NAR using a curriculum for the attention mask on translation tasks with seq2seq labels. Additionally, Gloeckle et al. (2024) focus on training models to achieve better and faster multi-token predictions as they scale up. These NAR approaches provide compelling alternatives to traditional AR LLMs, yet few have thoroughly explored training large NAR models on large-scale unlabeled data.

## 6 CONCLUSION

Building on existing DLMs, we present a recipe for building DLMs by continuing training on off-the-shelf autoregressive LLMs. Our adaptation technique involves using 1) attention mask annealing to enable bidirectional modeling and 2) shift operation to allow similar training dynamics like AR models. By unifying the language modeling objectives of autoregressive and diffusion models, we train diffusion models up to 7B parameters. Through experiments on common sense reasoning, language modeling, math reasoning and code generation, we show that DiffuGPT and DiffuLLaMA have better performance compared to existing DLMs. We find that DiffuLLaMA is capable of following in-context demonstrations to some extent on math problems. In the future, we aim to instruction tune our DLMs and explore inference time planning methods. We release DiffuLLaMA and DiffuGPT for further exploration of diffusion models as an alternative language modeling method.

AUTHOR CONTRIBUTIONS

Shansan Gong: Project lead, methodology development, DiffuGPT training and model evaluation, major writing. Shivam Agarwal: Methodology exploration, discussion, DiffuLLaMA training, writing. Yizhe Zhang: Discussion, DiffuLLaMA training, writing suggestions. Jiacheng Ye: Initial methodology exploration. Lin Zheng: Discussion, writing. Mukai Li & Chenxin An: Discussion, writing suggestions. Others: Mentorship and supervision.

ACKNOWLEDGMENTS

Research was supported in part by US DARPA INCAS Program No. HR0011-21-C0165 and BRIES Program No. HR0011-24-3-0325, National Science Foundation IIS-19-56151, the Molecule Maker Lab Institute: An AI Research Institutes program supported by NSF under Award No. 2019897, and the Institute for Geospatial Understanding through an Integrative Discovery Environment (I-GUIDE) by NSF under Award No. 2118329. This work used Delta AI at University of Illinois Urbana-Champaign through allocation CIS230229, CIS240488 from the Advanced Cyberinfrastructure Coordination Ecosystem: Services & Support (ACCESS) program, which is supported by U.S. National Science Foundation grants #2138259, #2138286, #2138307, #2137603, and #2138296.

This research was supported in part by the joint research scheme of the National Natural Science Foundation of China (NSFC) and the Research Grants Council (RGC) under grant number N_HKU714/21.

This work was also in part supported by research awards from Apple and the Allen Institute for AI.

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

# A  OBJECTIVE DERIVATIONS

This section provides detailed preliminary and loss derivations of §2 and §3.1 in the main paper.

## A.1  BACKGROUND OF DIFFUSION MODELS

We denote $\mathbf{x}_0 \sim p_{data}(\mathbf{x}_0)$ as the variable following the data distribution, and $\mathbf{x}_t \sim q(\mathbf{x}_t)$ as the noisy variable of $\mathbf{x}_0$ at time $t$, where the maximum time is $T$. The forward process

$$q(\mathbf{x}_{1:T}|\mathbf{x}_0) = \prod_{t=1}^{T} q(\mathbf{x}_t|\mathbf{x}_{t-1}) \tag{9}$$

corrupts the initial data $\mathbf{x}_0$ into a sequence of increasingly noisy variables $\mathbf{x}_{1:T}$. Accordingly, the reverse Markov process models the joint probability as

$$p_\theta(\mathbf{x}_{0:T}) = p_\theta(\mathbf{x}_T) \prod_{t=1}^{T} p_\theta(\mathbf{x}_{t-1}|\mathbf{x}_t), \tag{10}$$

which gradually denoises $\mathbf{x}_t$ to reconstruct the original data $\mathbf{x}_0$. Parameters $\theta$ are learned by minimizing the negative log-likelihood of $\mathbf{x}_0$, which can be optimized through the variational lower bound (VLB):

$$-\log p_\theta(\mathbf{x}_0) \leq \mathbb{E}_{q(\mathbf{x}_1|\mathbf{x}_0)}[-\log p_\theta(\mathbf{x}_0|\mathbf{x}_1)] + D_{\mathrm{KL}}(q(\mathbf{x}_T|\mathbf{x}_0)||p_\theta(\mathbf{x}_T)) + \mathcal{L}_T, \tag{11}$$

$$\text{with } \mathcal{L}_T = \sum_{t=2}^{T} \mathbb{E}_{q(\mathbf{x}_t|\mathbf{x}_0)}[D_{\mathrm{KL}}(q(\mathbf{x}_{t-1}|\mathbf{x}_t, \mathbf{x}_0)||p_\theta(\mathbf{x}_{t-1}|\mathbf{x}_t))]. \tag{12}$$

For continuous text diffusion (Li et al., 2022; Gong et al., 2023b), at each forward step, perturbations are applied according to

$$q(\mathbf{x}_t|\mathbf{x}_{t-1}) = \mathcal{N}(\mathbf{x}_t; \sqrt{1-\beta_t}\mathbf{x}_{t-1}, \beta_t \mathbf{I}), \tag{13}$$

where $\beta_t \in (0,1)$ represents different scales. In the end, $\mathbf{x}_T \sim \mathcal{N}(0, \mathbf{I})$. In the case of discrete denoising models (Ho et al., 2020; Austin et al., 2021; Zheng et al., 2024a), $\mathbf{x}_t$ follows a categorical distribution which naturally aligns with discrete text data. Let $\boldsymbol{x}$ be the one-hot encoded sample of variable $\mathbf{x}$ and $\mathbf{x}_t \sim \mathrm{Cat}(\boldsymbol{x}_t; \boldsymbol{p})$ represent a categorical distribution over vector $\boldsymbol{x}$ with probabilities given by $\boldsymbol{p}$. Here, $K$ represents the vocabulary size, $\boldsymbol{x} \in \{\boldsymbol{e}_1, \ldots, \boldsymbol{e}_K\}$, and $\boldsymbol{e}_k \in \{0,1\}^K$ is the one-hot encoding of the $k$-th word category. The forward process can be formulated through a transition matrix $\boldsymbol{Q}_t \in [0,1]^{K \times K}$ such that

$$q(\mathbf{x}_t|\mathbf{x}_{t-1}) = \mathrm{Cat}(\boldsymbol{x}_t; \boldsymbol{Q}_t^\top \boldsymbol{x}_{t-1}); \boldsymbol{Q}_t = (1-\beta_t)I + \beta_t \mathbf{1}\boldsymbol{e}_K^\top, \tag{14}$$

with $\mathbf{1}$ as an all-one vector of size $K$ and we assume $\boldsymbol{e}_K$ as the special [mask] state, also defined as the absorbing state in discrete diffusion $\boldsymbol{m}$. Each entry in $[\boldsymbol{Q}_t]_{ij}$ denotes the probability of transition from the state $\boldsymbol{e}_i$ to $\boldsymbol{e}_j$, and thus the previously defined $\boldsymbol{Q}_t$ means with probability $1 - \beta_t$, $\boldsymbol{x}_t$ will stay unchanged and otherwise it will jump to the mask state $\boldsymbol{e}_K$.

Starting from $\boldsymbol{x}_0$, the $t$-step marginal distribution and the posterior at previous time $t-1$ is respectively

$$q(\boldsymbol{x}_t|\boldsymbol{x}_0) = \mathrm{Cat}(\boldsymbol{x}_t; \boldsymbol{p} = \overline{\boldsymbol{Q}}_t^\top \boldsymbol{x}_0); \ q(\boldsymbol{x}_{t-1}|\boldsymbol{x}_t, \boldsymbol{x}_0) = \frac{q(\boldsymbol{x}_t|\boldsymbol{x}_{t-1}, \boldsymbol{x}_0)q(\boldsymbol{x}_{t-1}|\boldsymbol{x}_0)}{q(\boldsymbol{x}_t|\boldsymbol{x}_0)} \tag{15}$$

where cumulative products $\overline{\boldsymbol{Q}}_t = \prod_{i=1}^{t} \boldsymbol{Q}_i = \alpha_t I + (1-\alpha_t)\mathbf{1}\boldsymbol{m}^\top$, and $\alpha_t = \prod_{i=1}^{t}(1-\beta_t)$. We expect $\alpha_T$ approaches 0 such that the full noise data $\boldsymbol{x}_T$ is equal to $\boldsymbol{e}_K$ with probability 1. In the following sections, we primarily takes the discrete diffusion formulation.

## A.2  LOSS DERIVATION

Previous discrete time $t \in [0, T]$ restricts $\boldsymbol{x}_t$ to fixed time points whereas the continuous-time sampling allows for more flexibility covering any point in the range (Kingma et al., 2021; Shi et al.,

2024; Zhao et al., 2024; Ou et al., 2024). In this case, $t$ runs from 0 to 1, corresponding to dividing $[0, 1]$ into $T$ intervals and let $T \to \infty$. For any two arbitrary time points, $0 \le s < t \le 1$, the forward modeling can be generalized from $q(\mathbf{x}_t|\mathbf{x}_{t-1})$ to $q(\mathbf{x}_t|\mathbf{x}_s)$. We uniformly adopt the notation of continuous-time in following sections.

Following the previous definition, after simplification, we have $q(\boldsymbol{x}_t|\boldsymbol{x}_0) = \alpha_t \boldsymbol{x}_0 + (1-\alpha_t)\boldsymbol{m}$, referring to the probability of transition to absorbing mask state. Given $q(\boldsymbol{x}_t|\boldsymbol{x}_0) = q(\boldsymbol{x}_t|\boldsymbol{x}_s)q(\boldsymbol{x}_s|\boldsymbol{x}_0)$, we can derive the transition distribution between two arbitrary times $s$ and $t$:

$$q(\boldsymbol{x}_t|\boldsymbol{x}_s) = \mathrm{Cat}(\boldsymbol{x}_t; \overline{\boldsymbol{Q}}_{s|t}^\top \boldsymbol{x}_s), \text{ with } \overline{\boldsymbol{Q}}_{s|t} = \overline{\boldsymbol{Q}}_s^{-1}\overline{\boldsymbol{Q}}_t = \frac{\alpha_t}{\alpha_s}I + (1 - \frac{\alpha_t}{\alpha_s})\mathbf{1}\boldsymbol{m}^\top. \tag{16}$$

Similarly, after simplification,

$$q(\boldsymbol{x}_t|\boldsymbol{x}_s) = \frac{\alpha_t}{\alpha_s}\boldsymbol{x}_s + (1 - \frac{\alpha_t}{\alpha_s})\boldsymbol{m}. \tag{17}$$

Following Zheng et al. (2024a); Shi et al. (2024) and extend the formulation to continuous time, we have the backward transition probability:

$$q(\boldsymbol{x}_s|\boldsymbol{x}_t, \boldsymbol{x}_0) = \frac{q(\boldsymbol{x}_t|\boldsymbol{x}_s)q(\boldsymbol{x}_s|\boldsymbol{x}_0)}{q(\boldsymbol{x}_t|\boldsymbol{x}_0)} = \begin{cases} \frac{1 \cdot (1-\alpha_s)}{1-\alpha_t} = \frac{1-\alpha_s}{1-\alpha_t} = 1 - \frac{\alpha_s - \alpha_t}{1-\alpha_t} & \text{if } \boldsymbol{x}_t = \boldsymbol{x}_s = \boldsymbol{m}, \\ \frac{(1-\frac{\alpha_t}{\alpha_s}) \cdot \alpha_s}{1-\alpha_t} = \frac{\alpha_s - \alpha_t}{1-\alpha_t} & \text{if } \boldsymbol{x}_t = \boldsymbol{m} \ne \boldsymbol{x}_s. \end{cases} \tag{18}$$

For $\boldsymbol{x}_t \ne \boldsymbol{m}$, the $q(\boldsymbol{x}_s|\boldsymbol{x}_t, \boldsymbol{x}_0)$ will stick to the observed data. For $\boldsymbol{x}_t = \boldsymbol{m}$, we get the simplified

$$q(\boldsymbol{x}_s|\boldsymbol{x}_t, \boldsymbol{x}_0) = \frac{\alpha_s - \alpha_t}{1-\alpha_t}\boldsymbol{x}_0 + \frac{1-\alpha_s}{1-\alpha_t}\boldsymbol{m}. \tag{19}$$

In diffusion process, the generative model aims to approximate the reverse transitions using a denoising model $p_\theta(\boldsymbol{x}_s|\boldsymbol{x}_t, f_\theta(\boldsymbol{x}_t)) \rightsquigarrow q(\boldsymbol{x}_s|\boldsymbol{x}_t, \boldsymbol{x}_0)$, where $f_\theta(\boldsymbol{x}_t)$ represents the probability vector obtained from the softmax applied to the logits generated by the neural network, usually using transformer networks (Vaswani et al., 2017) in text domain. We can similarly have

$$p_\theta(\boldsymbol{x}_s|\boldsymbol{x}_t) = \frac{\alpha_s - \alpha_t}{1-\alpha_t}f_\theta(\boldsymbol{x}_t) + \frac{1-\alpha_s}{1-\alpha_t}\boldsymbol{m}. \tag{20}$$

Given Eq.19 and Eq.20, the KL-divergence loss is optimized by

$$D_{\mathrm{KL}}(q(\boldsymbol{x}_s|\boldsymbol{x}_t, \boldsymbol{x}_0)||p_\theta(\boldsymbol{x}_s||\boldsymbol{x}_t)) = \begin{cases} \frac{\alpha_s - \alpha_t}{1-\alpha_t}D_{\mathrm{KL}}(\boldsymbol{x}_0||f_\theta(\boldsymbol{x}_t)), & \text{for } \boldsymbol{x}_t = \boldsymbol{m}; \\ 0, & \text{for } \boldsymbol{x}_t \ne \boldsymbol{m}. \end{cases} \tag{21}$$

We can use the indicator function $\delta_{\boldsymbol{x}_t, \boldsymbol{m}}$ to unify the conditional cases. In addition, given $\boldsymbol{x}_0$, we have $D_{\mathrm{KL}}(\boldsymbol{x}_0||f_\theta(\boldsymbol{x}_t)) = -\boldsymbol{x}_0^\top \log f_\theta(\boldsymbol{x}_t)$ which corresponds to the cross-entropy widely used in the classification. Therefore, we have

$$D_{\mathrm{KL}}(q(\boldsymbol{x}_s|\boldsymbol{x}_t, \boldsymbol{x}_0)||p_\theta(\boldsymbol{x}_s||\boldsymbol{x}_t)) = -\frac{\alpha_s - \alpha_t}{1-\alpha_t}\delta_{\boldsymbol{x}_t, \boldsymbol{m}}\boldsymbol{x}_0^\top \log f_\theta(\boldsymbol{x}_t). \tag{22}$$

Following Eq.12, if we set a small timestep $\Delta_t = t - s = \frac{1}{T} \in (0, 1)$,

$$\mathcal{L}_T = \sum_{t=2}^{T}[-\frac{\alpha_s - \alpha_t}{(t-s)(1-\alpha_t)}\delta_{\boldsymbol{x}_t, \boldsymbol{m}}\boldsymbol{x}_0^\top \log f_\theta(\boldsymbol{x}_t)\Delta_t]. \tag{23}$$

By taking the limit as $T \to \infty$, we have $\alpha_t' = \frac{\alpha_t - \alpha_s}{t-s}$, and the sum is transformed into an integral:

$$\lim_{T\to\infty} \mathcal{L}_T = \int_0^1 \frac{\alpha_t'}{1-\alpha_t}\mathbb{E}_{q(\mathbf{x}_t|\mathbf{x}_0)}[\delta_{\boldsymbol{x}_t, \boldsymbol{m}}\boldsymbol{x}_0^\top \log f_\theta(\boldsymbol{x}_t)]\, dt. \tag{24}$$

Also, the first two terms in Eq.11 are $\to 0$ and a constant, respectively. Thus we can formulate the evidence lower bound (ELBO) of $-\log p_\theta(\mathbf{x}_0)$ as Eq.24.

The same form of ELBO which is invariant to noise schedule but related to the signal-to-noise ratio (SNR) is also introduced in Kingma et al. (2021); Shi et al. (2024). Following Austin et al. (2021); Zheng et al. (2024a), we choose the noise schedule $\alpha_t = 1 - t$, then $\frac{-\alpha_t'}{1-\alpha_t} = \frac{1}{t}$.

The previous discussion focused on the single token $\boldsymbol{x}_t$, and can be easily extended to a text sequence of length $N$ represented as $\mathbf{x}_t = [\boldsymbol{x}_t^1, \boldsymbol{x}_t^2 \ldots, \boldsymbol{x}_t^N]$. The final loss of the whole sequence is

$$\mathcal{L}_t^{1:N} = \frac{1}{t}\mathbb{E}_{q(\mathbf{x}_t|\mathbf{x}_0)}\left[-\sum_{n=1}^{N}\delta_{\mathbf{x}_t^n,\boldsymbol{m}}(\mathbf{x}_0^n)^{\top}\log f_\theta(\mathbf{x}_t^{1:N})_n\right], \tag{25}$$

where $f_\theta(\mathbf{x}_t^{1:N})_n$ denotes the whole input sequence is fed into the transformer model and the $n$-th output token is indexed. During training, we sample $t$ for each data point to optimize the expectation in $\mathcal{L}_t^{1:N}$ instead of the integral $\mathcal{L}_T$, while for evaluation, we use integral $\mathcal{L}_T$.

## B  IMPLEMENTATION DETAILS

### B.1  TRAINING DATA

**DiffuGPT**  Previous diffusion language models such as Plaid 1B (Gulrajani & Hashimoto, 2023), SEDD (Lou et al., 2024) and MD4 (Shi et al., 2024) use OpenWebText (Gokaslan & Cohen, 2019) to pre-train from scratch, referring to GPT2 (Radford et al., 2019). We choose the advanced FineWeb[2] corpus (Penedo et al., 2024), which is also derived from Common Crawl. We randomly sample 30 billion tokens from subset `sample-100BT`.

**DiffuLLaMA**  Following Zhang et al. (2024b)[3] we construct the training data for DiffuLLaMA by mixing SlimPajama (Soboleva et al., 2023) and Starcoder data (Li et al., 2023a). We randomly sample 65 billion tokens in the ratio of 7:3 from SlimPajama and Starcoder, respectively. We use sequence packing and pre-tokenize the dataset for efficient computing.

**Data Selection Consideration**  Our dataset selection aims to align with their respective pre-training objectives. Since we train on a relatively smaller number of tokens and do not intend to introduce new capabilities, we prioritize maintaining continuity with the models' pre-training distributions to minimize distributional shift. For GPT2-based DiffuGPT, we use the FineWeb dataset, which closely resembles OpenWebText (the dataset used in GPT2's pre-training). Considering that SEDD (Lou et al., 2024) iterates OpenWebText for more than 1 epoch and the total training amount is around 200B tokens, we also iteratively train DiffuGPT on 30B FineWeb data to more than 100B training tokens. In contrast, LLaMA2 is pre-trained over 1T tokens within one epoch on web and code data. To align with this, we follow TinyLLaMA (Zhang et al., 2024a) and use a mixture of SlimPajama and Starcoder data, designed to reflect LLaMA2's pre-training data.

### B.2  MODEL OPTIMIZATION AND HYPERPARAMETERS

**DiffuGPT**  We implement DiffuGPT using LLaMA-Factory[4] with DeepSpeed Zero-2 parallelization (Rajbhandari et al., 2020). The hyperparameter setting compared with previous work is listed in Table 4. The global batch size is calculated by multiplying the single GPU batch size, the number of gradient accumulation steps, and the number of GPUs, where we use 8 A100 80G. We use learning rate of $3e-4$ with cosine scheduler. The warm up steps are set to 2K and attention mask annealing steps are 10K. As shown in Table 4, our effective training tokens are less or equal than SEDD and MD4, while DiffuGPT exhibits better performance according to Table 1 and Figure 3.

**DiffuLLaMA**  For a more efficient pre-training, we implement DiffuLLaMA using huggingface[5]. We use DeepSpeed Zero-3 parallelization with CPU offloading (Rajbhandari et al., 2020) to efficiently scale DiffuLLaMA to multiple GPUs and nodes. Furthermore, we use flash-attention 2 and fused cross-entropy loss for optimized GPU memory usage and compute time (Dao, 2024). With these settings, we get set a batch size of 60 per GPU with context length 2048 on a GH200 96GB GPU. We use AdamW (Loshchilov & Hutter, 2019) to optimize our models with a constant learning rate of $2e-5$ and accumulate gradients every 4 steps. We train our model for 65 billion tokens on 16 4xGH200 nodes.

---

[2]https://huggingface.co/datasets/HuggingFaceFW/fineweb
[3]https://github.com/jzhang38/TinyLlama
[4]https://github.com/hiyouga/LLaMA-Factory
[5]https://github.com/huggingface/transformers

Table 4: Training settings for different diffusion language models.

| Models | Training steps | Global batch size | Context length |
|---|---|---|---|
| SEDD (Lou et al., 2024) | 400k | 512 | 1024 |
| MD4 (Shi et al., 2024) | 1000k | 512 | 1024 |
| DiffuGPT-S | 1000k | 256 | 512 |
| DiffuGPT-M | 160k | 1280 | 1024 |

**Tokenizer**   During adaptation, we do not change the tokenizer of the base model. In theory, we should expand the original vocabulary by adding an additional dimension to include a special token as `[MASK]` token. However, considering practical issues on implementation, we can alternatively select an existing word from the vocabulary to serve as the `[MASK]` token. It is preferable that this chosen word has a particularly low frequency of occurrence in corpus. For DiffuGPT-S we use `tokenid=10541` and for DiffuGPT-M we set a new `[MASK]` token with `tokenid=50257`. For DiffuLLaMA, we set `tokenid=811`.

### B.3   EVALUATION DETAILS

**Generation tasks**   For the TriviaQA and Lambada sentence completion tasks, we generate $n$-tokens for continue-writing. In triviaQA, we set $n$ to the oracle length plus an additional 10 tokens, and we only evaluate the first 2000 cases in this dataset for efficiency. For Lambada, which requires the completion of the last word, we set $n$ to oracle length of that word's tokens, which might be larger than 1 based on the tokenizer. For DLMs, we set the diffusion timesteps $T$ to the required generation length. For AR baselines, we cut off maximum new tokens. For SATMATH and MAWPS, we integrate our model into `math-evaluation-harness`[6].

**CommonSense Reasoning tasks**   The 4 commonSense reasoning tasks are multiple-choices questions with 4 options. Instead of open generation, we calculate the diffusion loss for each `Question+choice` pair using Eq.25. A lower loss (perplexity) indicates the model thinks that choice most suitable. This approach is commonly employed in ICL of LLMs (Wu et al., 2023). We also use this for AR baselines.

**Finetune GSM8K-symbolic**   The setting of finetune GSM8K-symbolic dataset is following Ye et al. (2024b)[7], which enables the diffusion model to perform chain-of-thought reasoning. For DiffuLLaMA, we use parameter-efficient-finetune: LoRA Tuning (Hu et al., 2022). We set rank to 8 and enable the finetuning of the word embedding layer, with 151 million (2%) parameters involved. For this task, we use $T = 64$ for the decoding of DLMs.

**Infilling tasks**   For ROCstories, where each case is a 5-sentence story, we setup evaluation referring Shen et al. (2023).The model is tasked with infilling the third sentence based on the first two and last two sentences. We evaluate the first 1000 cases in this dataset for efficiency. For code infilling, we use humaneval-single-line infilling [8] and their evaluation toolkit, which contains 1033 test cases. We implement infilling tasks for AR models by feeding the prefix and cutting off the generation length using the oracle length, considering that these AR models are not supporting infilling. We also try to feed the suffix information using the instruction like `Given prefix and suffix please infill the middle`, however, LLaMA2 can not follow this instruction. For AR LLMs, to perform infilling tasks requires additional FIM training (Roziere et al., 2023) or carefully instruction tuning.

**Unconditional Generation**   For unconditional generation in Figure 3, we set the temperature of top-$k$ to 0.98 and top-p to 0.9 for the medium-sized model, while using top-$k$ of 1.0 and top-p of 0.9 for the small model. We generate 64 samples and evaluate the perplexity using the GPT-2 large model, aligning with Lou et al. (2024); Shi et al. (2024).

---

[6] `https://github.com/ZubinGou/math-evaluation-harness`
[7] `https://github.com/HKUNLP/diffusion-of-thoughts`
[8] `https://github.com/openai/human-eval-infilling`

## C    ADDITIONAL RESULTS

### C.1    UNCONDITIONAL GENERATION

The generation quality is different for different hyperparameters, shown in Figure 5. Lowering the temperature increases fluency but reduces diversity, leading to noticeable repetition in sentences.

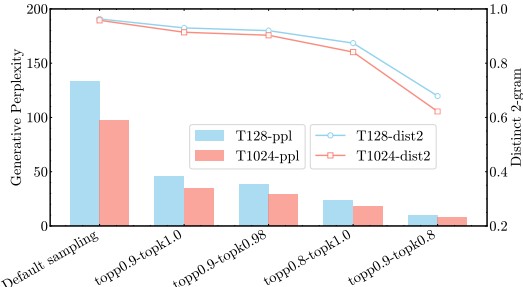

Figure 5: The unconditional generation quality for different diffusion time steps $T$ and sampling algorithms. We annotate the temperature of top-$k$ sampling and top-p sampling.

We randomly selected samples generated by our DiffuGPT-M models with 1024 tokens for various $T$, as shown in Table 12, Table 13, and Table 14. Lower $T$ values result in less fluent text.

### C.2    ABLATION ON GSM8K-SYMBOLIC

Using the same discrete diffusion loss, if we direct finetune on GPT2 achieves accuracy of 45.4 and 49.7 for small and medium models, respectively. In contrast, Finetuning from DiffuGPT yields accuracy of 50.2 and 61.8 (Table 1). Comparing with GPT2, DiffuGPT, as the base model, converges faster and attains a lower loss, as shown in Figure 6. This indicates that a better base model leads to improved results and also demonstrates the superiority of DiffuGPT as the current best diffusion base model.

Training with the initial weightings of GPT2, we evaluate three loss functions: DD, DD (no shift), and DD (no annealing) when finetuning on GSM8K-symbolic data. The corresponding loss and accuracy are shown in Table 5. Additionally, results for DiffuGPT and DiffuLLaMa are presented. All results highlight the negative correlation between the loss and accuracy.

Table 5: The training loss (ELBO) and test accuracy on the GSM8K-symbolic dataset.

| Models | Loss (ELBO) | Acc |
|---|---|---|
| GPT2-M + DD | 0.015 | 49.7 |
| GPT2-M + DD (no shift) | 0.028 | 34.5 |
| GPT2-M + DD (no anneal) | 0.019 | 47.2 |
| DiffuGPT | 0.009 | 61.8 |
| DiffuLLaMA | 0.003 | 63.1 |

### C.3    ADVANTAGES OF DLMS

To explore the self-correction advantages of DLMs noted by Ye et al. (2024b), we perform a qualitative analysis and find a similar self-correction capability in DiffuGPT. Our observation of the final steps of sampling trajectories, as shown in Table 6, indicates that DLMs refine intermediate numbers without adhering to a left-to-right constraint.

For global planning, we follow Ye et al. (2024a) to finetune DLMs on counting down (CD) datasets. CD is a mathematical reasoning challenge and a generalized version of the game 24, which many AR models struggle with (Gandhi et al., 2024). We compare DiffuGPT with other AR baselines with different model sizes in Table 7, demonstrating the advantages of DLMs.

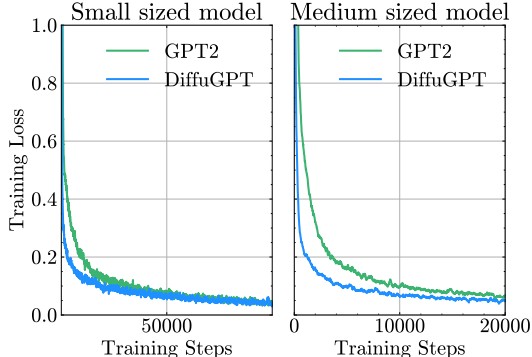

Figure 6: Finetune GSM8K data with discrete diffusion objectives, using a base model of either GPT2-S/M or DiffuGPT-S/M. DiffuGPT converges faster and attains a lower loss.

Table 6: A Case study to show self-correction capacity of DiffuGPT. $t/T$ refers to the current decoding step over the total diffusion steps. The incorrect rationales are marked in red.

| Steps ($t/T$) | DoT rationales |
|---|---|
| ... | ... |
| 9/32 | `<<3*15=45>> <<4*45=180>> <<180+300=00>> #### 00` |
| 8/32 | `<<3*15=45>> <<4*45=180>> <<180+400=580000 #### #### 000` |
| 7/32 | `<<3*15=45>> <<4*45=180>> <<180+400=400>> #### 480` |
| 6/32 | `<<3*15=45>> <<4*45=180>> <<180+300=500>> #### 580` |
| 5/32 | `<<3*15=45>> <<4*45=180>> <<180+300=580>> #### 580` |
| 4/32 | `<<3*15=45>> <<4*45=180>> <<180+300=480>> #### 580` |
| 3/32 | `<<3*15=45>> <<4*45=180>> <<180+300=480>> #### 580` |
| 2/32 | `<<3*15=45>> <<4*45=180>> <<180+300=480>> #### 480` |
| 1/32 | `<<3*15=45>> <<4*45=180>> <<180+300=480>> #### 480` |

For infilling tasks, we attempt to query the LLaMA model with the prompt `given the <prefix> and <suffix>, please answer the <middle> part`, which includes both prefix and suffix information. However, this approach is no better than simply completing the prefix, likely because the LLaMA model needs tuning for filling in the middle (FIM; Bavarian et al. 2022b). Additionally, Bavarian et al. (2022b) notes that using AR models for infilling presents challenges, such as prompting difficulties and repetition. In contrast, DLMs are naturally suited for this task, as they are trained to handle masked inputs, which is a key advantage.

Additionally, we conduct a controlled experiment by training both AR and DLMs on 100M tokens from the Starcoder dataset, using CodeLLaMA as the base model and evaluating performance on HumanEval infilling. We finetune CodeLLaMA autoregressively with FIM in both suffix-prefix-middle (SPM) and prefix-suffix-middle (PSM) formats. Our results in Table 8 show that Diffu-CodeLLaMA outperforms PSM, suggesting that prompt format affects AR models but not DLMs. We believe that training on more than 100M tokens, which is relatively small, could enhance performance.

Table 7: The finetuning results (accuracy) on the CD4 dataset.

| Models | Size | CD4 |
|---|---|---|
| GPT2-scratch | 85M | 45.8 |
| LLaMA FT | 13B | 51.1 |
| SoS (Gandhi et al., 2024) | 250M | 54.2 |
| DiffuGPT | 355M | 87.5 |

Table 8: Models finetuned on 100M tokens of Starcoder and their results on HumanEval Infilling.

| Models | Pass@1 HumanEval Infilling |
|---|---|
| CodeLLaMA FT (FIM-SPM) | 0.80 |
| CodeLLaMA FT (FIM-PSM) | 0.74 |
| Diffu-CodeLLaMA (Ours) | 0.76 |

### C.4 CONTINUAL PRE-TRAINING AR MODELS

We conduct a continual pre-training of GPT2 on the same corpus under the same settings as DiffuGPT. However, the zero-shot performance, shown in Table 9, indicates no improvement. This may be due to the stability gap introduced by continual pre-training (Guo et al., 2024), leading to performance degradation. Additionally, since our used corpus is similar to the one used for GPT2's initial pre-training, continual pre-training may offer limited new knowledge.

Table 9: Performance of different models on various tasks.

| Models | HSwag | Wino | SIQA | PIQA | Code |
|---|---|---|---|---|---|
| GPT2-M | 38.3 | 50.7 | 37.7 | 67.4 | 2.6 |
| GPT2-M (continue pretrain) | 36.7 | 49.4 | 37.9 | 66.7 | 2.6 |
| DiffuGPT-M | 37.2 | 52.6 | 39.0 | 59.6 | 2.9 |

### C.5 DECODING SPEED TESTING

We evaluate the inference time of LLaMA2 and DiffuLLaMA for unconditional text generation across various lengths. Our tests include vanilla attention, flash attention 2, and the torch version of flash attention SDPA, as shown in Table 10.

Table 10: Single batch inference time for different attention implementation and generation lengths.

| Length | Attention | DiffuLLaMA (sec) | LLaMA (sec) |
|---|---|---|---|
| 512 | flash-attention 2 | 12.5 | 9.2 |
| 1024 | SDPA | 13.2 | 16.3 |
| 1024 | flash-attention 2 | 13.3 | 17.5 |
| 1024 | vanilla | 16.2 | 17.2 |
| 2048 | SDPA | 28.5 | 29.5 |
| 2048 | flash-attention 2 | 23.5 | 35.7 |
| 2048 | vanilla | 38.1 | 32.8 |

Yet smaller $T$ leads to faster generation, in downstream tasks like multiple-choices, this may slightly impact accuracy. Examples are provided in Table 11.

Table 11: Performance of DiffuLLaMA models on various tasks.

| Models | HSwag | Wino | SIQA | PIQA |
|---|---|---|---|---|
| DiffuLLaMA T=32 | 58.7 | 56.4 | 43.2 | 63.3 |
| DiffuLLaMA T=8 | 47.1 | 52.6 | 41.9 | 57.1 |

Table 12: Generation examples of DiffuGPT-M ($T = 1024$).

If you're considering applying to the JBCC school, I'm confident you will find something there!
What exactly do the schools have required?
I HAVE DREAMED (I know it's not in my DNA). In fact, I have my life time.
What do you need anyway?
- MA or PhD degree in non-religious education and/or MA or PhD degree.
- Our Schools ensure that you have a strong academic background in the non-religious field of your choice and a passion for teaching in literature, research, writing, reading, English as a second language, or teaching.
- The concentration may help to provide a world-class emphasis or offer unique opportunities for teaching of new skills or in dynamic contexts.
- Practical Program coursework enables students to study in specific fields and areas in their career, this may be an essential part of education if you work as a tutor or if your goal will include working in an office, as a high school social science teacher, or a classroom.
What makes JBCC unique?
The National International Baccala Practical Programs are located locally and offer JBCC's scope and the preparation to complete an advanced master's degree.
How do students go to graduate from JBCC?
Get in now!
Congratulations for applying for the JBCC program. Become a part of this community.
What does JBCC achieve?
JBCC prepares leaders to be a catalyst for an educational and emotional enriching environment for service. Leaders are within their capacity to call others to service. Want to know more?
To find additional information, please use this form.
You can reach us to find your application materials for JBCC here.
Christine Callender
Educator and Director

School of Christian Education Hello there everyone! Welcome to our tutorials section. We have everything you need to know what and how to make quality t-shirts apparel especially for those who are new to t-shirts. If you are not sure, they are a very popular item.
As you might imagine, there are a wider audience of people than others.
- T- t shirt – how to make yourself?
- T-shirt to wear?
- What can I use to make my t shirt?
...
4. Do I need to reuse my tee shirt?
There is no need to reuse your tee shirt. This means you can spend a few minutes putting it on another pair of clothing. Polyester is great for quick clean, allowing you to get out the stains on the front of your tee shirt, without damaging them.
Since polyester is very breathable, which means it is able to be used on both your skin and anything else

Table 13: Generation examples of DiffuGPT-M ($T = 256$).

Use of the Service: Cookies may enable your internet site or your computer or device to access information using our Services, so as to allow us to work together to provide the website that you use. Cookies may provide Personal Information in certain cases.
Google Digital Advertising Cookies
Advertising partners may use internet analytics with information on your websites visit, how far you come, through our different advertising networks in order to tailor them for you. This may provide your personal information to our advertising partners and may be shared with third-party advertisers.
Google may also use tracking cookies that analyse visitor preferences, such as the type of device used, visits to pages the user has access to most frequently, or how frequently visitors visit particular pages, to analyse how people find what sort of information most interest to them. They can keep track of all such visits as they collect more information, using cookies and analytics to improve the content of our website.
For more information on how to control Google's cookies on our websites, see https://geo.com/sies to learn moreLogan has been very popular among travelers in the area since 1845. The early explorers had a place and that's the best place to stay after a long journey. Fast-forward the years, and the ones that stayed in Logan migrated before even crossing over into the mountains to the north. These changes have made it from a midwestern city to a postmodern city of sorts where everyone lives.
This is what global locations mean. It's easy enough to get lost after a day spent traveling, but every city offers different experiences and fun when it comes to being on the road.
The most important parts of Logan experience are the historic districts of Logan Square. Discover the history of so many different places and gather with people who enjoy exploring something new, or even if with children or pets in tow. Discover the Broadway complex of Logan Square and the rest of the neighborhood, as it's a charm sprinkled with charm.
Sitting the North of Telegraph Hill one mile from Loomers you will find the Logan mountain trail. The 18-mile-long trail is one of the finest in the state and enjoy a day on your bike or biking, surrounded by all sorts of cafes, unique eateries and art galleries.
Lawsony Park has the Home Park Beer Cellar and the Home Park Bike Shop, and a great gift shop for that. Explore the neighborhood on Memorial weekend and warm up with a delicious bite from Logan Friendly Brews.
A week on Memorial Weekend is a new tradition for the area of Loomers. Here's the excitement of heading on the road on one of the weekends of the year!
The summer at Logan is one of the most popular times of year to explore the quirky neighborhoods. Our parks are vibrant and busy. We are downtown and our restaurant and bars stay open to have a drink and coffee. Stop in at the rooftop patio to enjoy a really nice evening with a quick bite. And every week on Memorial Weekend, you'll find a really nice indoor pool!
For kids with a weekend, Logan has a playground and good for two and on August 4 and 8 is the Logan Fun Day. A few spots are available (and we're always open to both men and women) and your group allows you to just catch up and season with friends while enjoying a pint of beer.
John Moody Brews is about to bring their Labor Day fun to the streets and your backyard on Labor Day Weekend. Logan Friendly Brews has a brewery cruise, street vendors and live music on Thursday night and they'll have giveaways on Sunday. It's a fun

Table 14: Generation examples of DiffuGPT-M in 1024 tokens ($T = 32$).

t work in near-zero conditions.

In the meantime, employees are planning to avoid travel, and lack of traveling. The United States will not pay or pay for travel, Grand Tech said. But the company plans to continue to take its staff to Asia and other destinations. The plans for the next trip will vary. The Philippine office is investigating the outbreak and is considering further expanding to more but still non-fue areas. There may also be more traveling than they had previously been doing before.

The company reports, confirmed first by ETOY's Sulayo Suan, denied to local media. Grand Tech has its headquarters in Cincinnati, Ohio. Delta variant virus caused a plant disaster nearly two months ago. The the genotype 3 virus of the virus hit an employee while on the job there in April.

The woman reported that she needed to remain home confinement until August. But succumbed continued serious illness to test results for weeks.

The Delta variant is currently targeting companies in the European Union, with the virus occurring in Italy. French workers have also been reported suffering from the outbreak. Employees have been traveling to work due to some journeys to Asia and respiratory complications to travel to Asia.

The company also reported significant risks to their herbal products containing Chinese plant material, including some hogweed. Hogweed alone is responsible for the loss of 1 U.S stem to each year, Grand Tech said.

The U.S. is offering an ongoing vaccination program available to 72 million all Americans, including those most at risk from the virus, State Street announced.Globular epilepsy is a genetic condition that is at risk.. ORIK can occur in as high as 15% of people living in Indonesia. However, I can discuss information that exists that ORIK.

There no factors that could be

MeanESK 2 is a class II agent with a teratogenic activity as determined by the MSL. We propose not treatment of ORIK on patients that exceed the 2.0 threshold.

Both biological factors predispose risk of the.

ORISA-WEVINB8: X at leastISMRC IX.

Non-Mouse type 8a(d12).

About both factors that promote the development of ORIK.

To specify a reasonable regulation that would affect a p.ss.The United States Juvenile Court as an Authority for Administration of the Bureau of Corrections by Arthur Whyte, Jr. Chair: John F. Bronz Board: Buck Morgan Jr. Rep: John Little Reader: David Gervis.

THURY Fisher, WOOD and her son.

DATE: March 23, 1959

REV: July 3, 1949

By resolution which is passed: Either a person uses nothing more than a boat within a warehouse, apartment or barn. Department of the Interior or Department of Labor or system of it existed as a whole under the laws of Michigan three. through December one of such it would not. Each $100 person shall be fined and the division shall collect all damages of each two hundred dollars and dollars' the total of such amount and enter a. Rights reserved: also a trustee thereof may order the delinquent bonds. : If a person convicted if that has committed any act of attachment. Assisting duties shall deemed to have ceased within fifteen days, then the Commissioner to be appointed shall pay the district the sum of four percent of the value of the money recovered from these costs. Pursuant in this section shall be revoked without notice. Revocation: also the equivalent of instructions accompanied with. Fifteen days of the entry of such order. : The blight or place known to be in a person has an occupancy.k enables.

He can throw in an about picture. How he loves it so much he is the greater these online webit himself right from scuba dive and wishes to be removed by the government of America, that's a much sleazier story. From time social circles, the members of MoolahFast, a dating site focusing on the planet of different scuba divers,

