# OpenReview forum: "Scaling Diffusion Language Models via Adaptation from Autoregressive Models"
_ICLR.cc/2025/Conference — ICLR 2025 Poster_

### Official Review · Reviewer_87p8 · 2024-11-02

**Soundness:** 4
**Presentation:** 2
**Contribution:** 3
**Rating:** 8
**Confidence:** 4

**Summary:**

The paper addresses limitations in autoregressive (AR) models, particularly issues with global planning and intermediate token correction. In response, Diffusion Language Models (DLMs) were introduced but face scalability challenges due to the computational costs of training with only 400 billion tokens. This work proposes a novel training mechanism that redefines DLM objectives. The approach also addresses architectural limitations in AR models, using attention mask annealing and a shift operation to remove causal masking bias. Scaled models ranging from 127 million to 7 billion parameters, specifically DiffuGPT and DiffuLLaMA, demonstrate better or comparable performance to existing AR models. Additionally, the authors release open-source code for these models, providing valuable resources to the research community for faster inference speeds with minimal iterations.

**Strengths:**

- Well-written and easy to understand.
- Comprehensive evaluation on multiple tasks, including word completion, reading comprehension, commonsense reasoning, math problems, and coding.
- Ablation study conducted to emphasize the importance of shift operation and attention mask annealing.
- Mathematically grounded and novel approach.
- Demonstrates increased token diversity without compromising quality.

**Weaknesses:**

- Missing qualitative examples and human evaluation.
- No statistical significance testing across tasks.
- The current T is set to 32. It would help to see the quality performance across different T, since it was mentioned in section 4.5 that decreasing T leads to faster decoding but with loss of quality.

**Questions:**

The current T is set to 32. It would help to see the quality performance across different T, since it was mentioned in section 4.5 that decreasing T leads to faster decoding but with loss of quality.
Please include this.

---

> ### Author Response · Authors · 2024-11-22
> **Response to reviewer 87p8**
>
> Thank you for your time and valuable suggestions. We list our point-by-point responses and hope to address your concerns.
>
> > Weak1: qualitative examples and human evaluation
>
> We list the unconditional generation results in Appendix C, which shows better quality with larger decoding T. For human evaluation, thank you for this suggestion. Most of the tasks can be evaluated using automatic metrics, and human evaluation is expensive. We will consider conducting human evaluation for some generation tasks if resource permits.
>
> > Weak2: significance testing
>
> All experimental results involving random seed are averaged from the results of 3 separate trials, with a confidence interval of p < 0.01 based on t-test. Experimental results also reveal significant disparities in accuracy among different models/tasks.
>
> > Weak3: different T
>
> We discussed the unconditional generation quality change with different T in Fig3, where less T resulted in higher PPL. For downstream tasks like multiple-choices in Table1, smaller T might result in a slight accuracy drop, we list some examples the following.
>
> |Tasks| HSwag| Wino| SIQA |PIQA|
> |---|----|---|---|---|
> DiffuLLaMA T=32| 58.7| 56.4 |43.2| 63.3|
> DiffuLLaMA T=8| 47.1| 52.6 |41.9 |57.1|

---

> > ### Comment · Reviewer_87p8 · 2024-11-26
> > **Official Comment**
> >
> > The authors have addressed most of the comments (minor) but might need additional human judgments. Will keep the score the same.

---

### Official Review · Reviewer_rZH1 · 2024-11-04

**Soundness:** 3
**Presentation:** 3
**Contribution:** 3
**Rating:** 6
**Confidence:** 4

**Summary:**

This paper proposes to adapt pre-trained autoregressive models to build discrete diffusion models via continual pre-training. This way, they can convert AR to diffusion parameters by training on fewer than 200B tokens. Technically, they propose  (1) annealed attention masking, (2) inheriting the shift inductive bias from AR. The experiment section suggests that this diffusion model demonstrates strong performance in math, word lambada, and infilling, and ablation suggests that the shift inductive bias is crucial for the transfer.

**Strengths:**

1. The paper proposes to adapt AR pretraining parameters to Diffusion parameters, which solves a very important problem about improving the training efficiency of diffusion models.
2. The result seems solid, with a fair amount of improvement over previous diffusion models. But I still think there are some unresolved problems as shown in the weakness section.
3. The experiment seems pretty comprehensive, with a wide range of downstream task, generation quality, analysis, ablation, and inference efficiency.

**Weaknesses:**

Since the paper claimed to tackle the parameter transferability problem, there remain some unresolved scientific questions:
(1) suppose we have more compute, is this transfer still optimal? would this transfer converge to a worse local optimum than if we train from scratch for longer?
(2) The paper included ablation results on GSM8K performance, but more fundamentally, what's the loss in terms of PPL for the (without shift) and (without annealing) baselines? I think these are more important and more general-purpose to report than a specific downstream task. I saw the disclaimer that PPL may not directly reflect the correctness and it's only ELBO. I still think it's good to report ELBO on the test set in the paper.
(3) What's the number of flops for the training? 200B tokens is a bit abstract, especially 200B token for diffusion is different from 200B token for AR (due to lack of KV caching) and 200B token means different Flops across model sizes.

**Questions:**

see weaknesses.

---

> ### Author Response · Authors · 2024-11-22
> **Response to reviewer rZH1**
>
> We thank reviewer rZH1 for valuable suggestions. We are happy to know that you think our experiment is comprehensive and solid. We’d like to discuss the unresolved scientific questions with you.
>
> > Weakness 1 What will happen given more computation? Will it converge to local optimum than training from scratch?
>
> Thank you for raising this interesting question. The experiments in transferring GPT2 give us a positive signal that transferring from GPT2 using DD loss will bring us quite good DLMs (compared with SEDD and GPT2, noting that DiffuGPT trains less tokens than SEDD). However, this has not been proven in LLaMA due to the resource limitation. We believe it is an important direction that needs to be examined in the future of the DLM community but this is beyond our current paper. In our paper, **we aim to show some positive signals of 7B DLMs with the limited resources.** Based on this, we believe the community and us have confidence in the next steps of exploring larger DLMs.
>
> > Weakness 2 Loss in Table 3
>
> We agree that ELBO can serve as the reference. We list the loss curve of different base models in Fig 6. Initailized with GPT2-M, we evaluate three loss functions: DD, DD-no shift, and DD-no anneal for fine-tuning GSM data. The corresponding loss and accuracy are shown in the table below. Additionally, results for DiffuGPT and DiffuLLaMa are presented for reference. All results highlight the negative correlation between the loss and accuracy.
>
> |Models|	Loss|	Acc|
> |-----|-----|------|
> GPT2-M w/ DD|		0.015|	49.7|
> GPT2-M w/ DD-no shift|	0.028|	34.5|
> GPT2-M w/ DD-no anneal	|0.019|	47.2|
> DiffuGPT-M	|0.009|	61.8|
> DiffuLLaMA |	0.003|	63.1|
>
> > Weakness 3 FLOPs of training
>
> KV-cache is usually used during inference time for LLMs. During training, under the same setting, the theoretical computing FLOPs of LLM and DLM should be very close. We use utils in deepspeed to get the FLOPs of AR LLMs and DLMs: For DiffuLLaMA 256tflops, for LLaMA 255tflops per step per gpu. We train 6950 steps for DiffuLLaMA and the total FLOPs are 1.7*10^18, which refers to 15.7K GPU hours on H200.

---

> > ### Author Response · Authors · 2024-11-28
> >
> > Dear Reviewer rZH1,
> >
> > Thank you for your valuable feedback on our manuscript. Your insights are instrumental in enhancing the quality of our paper.
> > We look forward to your response and hope that our revisions have addressed your concerns effectively.

---

### Official Review · Reviewer_ZRNS · 2024-11-04

**Soundness:** 2
**Presentation:** 4
**Contribution:** 3
**Rating:** 5
**Confidence:** 3

**Summary:**

The paper introduces a methodology for adapting pretrained autoregressive (AR) language models to discrete diffusion (DD) models. It identifies two main points of conflict between them: 1) Causal attention (AR) vs. bidirectional attention (DD), 2) next-token-prediciton (AR) vs. denoising (DD). In order to resolve 1), the paper proposes attention mask annealing, where the causal attention mask is slowly turned into a full attention mask over training. For 2), the DD loss is adapted by shifting the logits by 1 to be in line with AR models.
In the experimental evaluation, they continue to finetune pretrained AR models of various sizes using the derived adaptation scheme. The models are evaluated on various knowledge, reasoning, coding, and math benchmarks.

**Strengths:**

* Adapting AR models to diffusion models is a problem that is of interest to the ICLR community.
* The proposed method is intuitive and simple.
* The paper is well written and flows really well.
* The results demonstrate that the adaptation is successful to some extent: Adapting larger LLMs leads to better performance.
* The DD model shows superior performance on some tasks that mostly require infilling.

**Weaknesses:**

1. In lines 364-368, it states that "by observing different scales of our adapted diffusion models, we can draw the conclusion that scaling diffusion language models results in improved performance". This statement suggests that if a DD model was trained from scratch, performance would scale with the size of the model. However, this isn't supported by the experiments, which merely shows that AR models of increasing size can be adapted to DD models in such a way, that larger adapted models perform better than smaller adapted models. However, it is possible that the adaptation process is merely retaining some of the knowledge/skills that were learned by the pretrained AR model. It doesn't mean that the DD model could attain these capabilities itself through scaling. I think it is important to be precise here.

2. In lines 367-369: "This discrepancy [between DiffuGPT's performance compared to the base model vs. DiffuLLaMA's performance compared to the base model] is presumably attributed to the extensive amount of training tokens processed by the DiffuGPT series models, whereas the 7B model has not undergone equivalent training." This explanation may be true, but needs to be verified. Since DiffuLLaMA has been pretrained on more than twice as many tokens (65B vs 30B), it doesn't seem obvious. What is the reason not to finetune both models on the same data?

3. In lines 373-375: "In tasks that require more extensive global reasoning, such as complex mathematics and coding, DLMs consistently exhibit better performance compared to AR models that rely solely on left-to-right modeling capabilities." This claim can only be substantiated through a controlled experiment that continues to finetune the base model on the same data in AR mode vs DD mode. Table 3 conducts this experiment, but only on a single downstream dataset that is quite small. Having a fair, controlled comparison between AR and DD models is the biggest weakness of this paper.

4. Table 2: Different models here correspond to different prompting methods. hit@k is not a prompting method, it is an evaluation metric whose value is at least as high as the standard evaluation metric with hit@1. Stylizing the respective row with hit@k incorrectly suggests that this is the best "method", which could mislead the reader into thinking that the difference to the LLaMA-2 baseline is smaller than it actually is.

**Questions:**

* From the paper, it is unclear to what extent the continuous-time discrete diffusion processes described in 3.1 are a novel contribution. If this is not novel, I would move it to section 2 for clarity about your contributions.

* In diffusion processes, do we need to know the number of tokens ahead of time in order to know the number of MASK tokens? If not, how is decoding done?

---

> ### Author Response · Authors · 2024-11-22
> **Response to reviewer ZRNS (1/2)**
>
> We thank reviewer ZRNS for the detailed review. We’re happy to know you find our method effective. We noticed your concerns are focused on certain statements in the experiment section. Below, we provide a point-by-point rebuttal to clarify any possible misunderstandings and revise our draft accordingly.
>
> > L364-368: "by observing different scales of our adapted diffusion models, we can draw the conclusion that scaling diffusion language models results in improved performance" This is not verified by training scratch.
>
> We believe there may have been a misunderstanding regarding our statement, as it was not intended to suggest "training from scratch". Please note that **our work does not discuss or focus on training diffusion models from scratch**.
>
> To clarify, our paper does not aim to investigate strict scaling laws, such as those observed in AR LLMs (e.g., openai scaling law [1]). Instead, we focus on adaptation and evaluating the adapted models. When we refer to “scaling diffusion models”, we are specifically discussing the performance of adapted models rather than models trained from scratch. It is possible that adaptation retains some of the knowledge and skills from AR LLMs, and the fact is we obtain a larger DLM (7B) which yields superior performance than smaller models. So the conclusion we draw is that larger adapted models perform better than smaller adapted models. In response to your feedback, we have revised lines [L364-L368](https://openreview.net/pdf?id=j1tSLYKwg8) to ensure our intention is clearer and to avoid any misinterpretation. We hope [this revision](https://openreview.net/pdf?id=j1tSLYKwg8) addresses your concern.
>
> > L367-369: "This discrepancy is presumably attributed to the extensive amount of training tokens processed by the DiffuGPT series models, whereas the 7B model has not undergone equivalent training." This is not verified.
>
> Considering the tendentious resources required for training 7B models, we stopped at 65B tokens, so we just assume the possible reason of DiffuLLaMA’s performance in this statement is the insufficient training.
>
> There might be misunderstandings because you think DiffuLLaMA has been pre-trained on more than twice as many tokens (65B vs 30B) than DiffuGPT but actually it is not correct. **65B vs 30B is the amount of data used, but not the amount of training tokens.** We should emphasize: (i) DiffuGPT training iterates more than one epoch on prepared 30B tokens (see the footnote in page 6) and in the end it consumes more than 100B tokens. DiffuLLaMA only iterates once using 65B tokens. (ii) 7B scale is much larger than GPT2-small and medium size, and according to scaling law [1] in AR LLMs, 65B tokens are definitely insufficient for 7B models.
>
> We rewrite [L367-369](https://openreview.net/pdf?id=j1tSLYKwg8) to avoid misunderstanding, please let us know if you have any additional questions.
>
> Regarding the training of DiffuGPT and DiffuLLaMA on different datasets, we mentioned this in Appendix (Line 1043) and provide further clarification here. **Our dataset selection aims to align with their respective pre-training objectives.** Since we train on a relatively smaller number of tokens and do not intend to introduce new capabilities, we prioritize maintaining continuity with the models' pre-training distributions to minimize distributional shift [2].
>
> For GPT2-based DiffuGPT, we use the FineWeb dataset, which closely resembles OpenWebText (the dataset used in GPT2's pre-training). Considering that SEDD [3] iterates OpenWebText for more than 1 epoch and the total training amount is around 200B tokens, we also iteratively train DiffuGPT on 30B FineWeb data to more than 100B training tokens. In contrast, LLaMA is pre-trained over 1T tokens within one epoch on web and code data. To align with this, we follow TinyLLaMA [4] and use a mixture of SlimPajama and Starcoder data, designed to reflect LLaMA 2's pre-training data.
> While we acknowledge that these dataset choices may result in performance differences, it is important to note the reasonable overlap between SlimPajama and FineWeb due to their similar data sources. We update this to [the dataset section of the appendix](https://openreview.net/pdf?id=j1tSLYKwg8) and the dataset mixtures for diffusion modeling could be explored in future work.

---

> > ### Comment · Reviewer_ZRNS · 2024-11-22
> >
> > Thank you for revising the language. I think the statements in the paper are correct now.

---

> ### Author Response · Authors · 2024-11-22
> **Response to reviewer ZRNS (2/2)**
>
> > L373-375: "In tasks that require more extensive global reasoning, such as complex mathematics and coding, DLMs consistently exhibit better performance compared to AR models that rely solely on left-to-right modeling capabilities." This is not compared with continuing pre-training AR.
>
> Thank you for highlighting the controlled comparison between AR and DD. We have added supporting experiments at the end.
>
> To clarify, lines L373-375 explain Table 1, while Table 3 presents a different experimental setup leading to distinct conclusions.
> In Table 3, we fine-tuned the GPT-2 model directly on GSM data using either AR or DD loss, without further training on the pre-training corpus. The superior performance of DD loss suggests that DD loss better leverages GSM data during fine-tuning.
> In Table 1, we obtain the adapted DLMs through continual pre-training on unsupervised pre-training corpus. We aim to evaluate their capacities in GSM fine-tuning and other zero-shot tasks like code infilling. **The goal here is not to compare how each loss utilizes the pre-training corpus, but rather to benchmark our models with other existing models (AR LMs vs. previous DLMs vs. our DLMs).** According to Table1, our DLMs are better than AR LMs in GSM and code. We update our [manuscript](https://openreview.net/pdf?id=j1tSLYKwg8) to avoid misunderstandings.
>
> We understand your concern regarding controlled comparisons with AR models continually pre-trained on the same corpus. To address this, **we conduct a continual pre-training of GPT2 on the same corpus under the same settings as DiffuGPT.** However, the loss curve flattens after 1k steps, and the zero-shot performance, shown below, indicates no improvement. As analyzed in paper [2], this may be due to the stability gap introduced by continual pre-training, leading to performance degradation. Additionally, since our used corpus is similar to the one used for GPT2’s initial pre-training, continual pre-training may offer limited new knowledge.
>
> |Tasks |HSwag| Wino |SIQA| PIQA |Code|
> |----|----|--|--|--|--|
> |GPT2-M |38.3 |50.7| 37.7| 67.4| 2.6|
> |GPT2-M (continue pretrain)| 36.7 |49.4| 37.9| 66.7| 2.6|
> |DiffuGPT-M| 37.2 |52.6| 39.0| 59.6| 2.9|
>
> > Unclear title in Table 2.
>
> Thank you for pointing this out. Actually, the first column is either different models or different prompting methods, because there are different models (DD/AR), different prompting methods (ZS/FS) and different evaluation methods (hit@k). So we will change “Models” to “Settings” here. Stylizing the respective row with hit@k into gray color does not indicate it is the best method, but just for the reference to show the model’s potential upper bound (mentioned in line 434), just like many papers will list the oracle results for reference in the table. We have added clarification to our [updated manuscript](https://openreview.net/pdf?id=j1tSLYKwg8).
>
> > Q1 The order of Section 3.1
>
> Thank you for pointing this out. Continuous time DD is recently discussed in other papers (see citations in the paper). It is close to our model so we list the details in section 3.1 as the preliminary of the whole section 3. We didn’t claim this as our contribution in the introduction. We will consider to move it to section 2.
>
> > Q2: Decoding length
>
> We need to pre-define the generation length ahead. We set a max number of new tokens by using a sequence of [MASK] tokens and split at the first [EOS] token to produce the final response. We do this post processing because it is a common practice in DLMs [5][6].
>
> **Reference**
>
> [1] Scaling Laws for Neural Language Models. OpenAI 2020
>
> [2] Efficient Continual Pre-training by Mitigating the Stability Gap. arxiv 2024
>
> [3] Discrete Diffusion Modeling by Estimating the Ratios of the Data Distribution. ICML 2024 Best paper
>
> [4] TinyLlama: An Open-Source Small Language Model. arxiv 2024
>
> [5] DiffuSeq: Sequence to Sequence Text Generation With Diffusion Models. ICLR 2023
>
> [6] A Reparameterized Discrete Diffusion Model for Text Generation. COLM 2024

---

> > ### Comment · Reviewer_ZRNS · 2024-11-22
> >
> > Thank you for the additional experiment and clarifications. I think this makes your paper better. I still think that it makes valuable contributions, like the adaptation method and some improved performance on mainly infilling tasks.
> >
> > However, my impression is still that the paper is trying to claim more credit than it deserves.
> > One of its core selling points is that it outperforms other existing DLM models. However, those models differ from the proposed ones in many ways, including how they were trained (from scratch rather than adapted). Comparing to these models that were trained from scratch and claiming SOTA among DLM models although you're "just" adapting existing AR models is not very meaningful in my opinion. The claim that you're able to scale DLM models is similarly a bit overclaiming because they are not trained from scratch. While the language was adapted in the specific passages that I pointed out, the paper still makes the impression in other places (e.g. first sentence of the conclusion).
> >
> > In summary, I raised my score to 5.

---

> > > ### Author Response · Authors · 2024-11-24
> > > **Response to reviewer ZRNS**
> > >
> > > Thank you for your thoughtful feedback and for acknowledging the improvements in our revised manuscript. It seems your main concern is whether adaptation-based models are meaningful, given they are not trained from scratch. We would like to emphasize the following points:
> > >
> > > 1. **Training from scratch vs. adaptation**: While training 7B DLM from scratch is undoubtedly meaningful, it is not contradictory to our work. Our decision to use adaptation was primarily driven by resource limitations, which are a practical concern in large-scale model development.
> > > 2. **Adaptation as a valid approach**: Many prior studies (e.g., [1][2]) investigating novel architectures like Mamba and RNN have also employed adaptation from existing large language models. This demonstrates that adaptation is a valid and widely-used approach for exploring new architectures/paradigms while minimizing computational costs.
> > > 3. **Future impact**: By adapting to produce a 7B DLM, we not only showed improved performance compared to smaller DLMs like SEDD but also made a larger DLM available for downstream testing. This significantly broadens the potential  study of DLMs in future, such as investing more resources into training 7B DLMs from scratch or exploring downstream tasks.
> > >
> > > **Reference**
> > >
> > > [1] The Mamba in the Llama: Distilling and Accelerating Hybrid Models. NeurIPS 2024
> > >
> > > [2] Gated Slot Attention for Efficient Linear-Time Sequence Modeling. NeurIPS 2024

---

> > > > ### Comment · Reviewer_ZRNS · 2024-11-25
> > > >
> > > > No. To be clear, I think the topic of your paper is perfectly valid. As noted in my previous comment, I however think that papers with a similar level of contribution do better in regards to clearly stating and supporting their contributions, although your presentation of the method is overall very good.

---

> > > > > ### Author Response · Authors · 2024-11-28
> > > > >
> > > > > Dear Reviewer ZRNS,
> > > > >
> > > > > Thank you very much for your constructive feedback. We truly appreciate your insights, which have significantly enhanced the quality of our paper. We are so encouraged by your acknowledgment of our work.
> > > > >
> > > > > In response to your latest feedback, we have revised all statements regarding "scaling" in the manuscript, hoping it can properly clarify our claims. We look forward to your response and hope to ensure that all your concerns have been addressed satisfactorily.
> > > > >
> > > > > Thank you once again for your valuable input.

---

> > > ### Author Response · Authors · 2024-11-25
> > > **Update and response to reviewer ZRNS**
> > >
> > > Regarding the potential *overclaim* about scaling DLM models, we would like to clarify a few points:
> > >
> > > We are not trying to overclaim that we trained DLMs from scratch. While a strict definition of "scaling" may indeed imply training from scratch, **our intention was to emphasize increasing the model size** to achieve capabilities comparable to existing LLMs. Given computational constraints, we chose adaptation as a practical approach to achieve this goal.
> > >
> > > To address possible confusion, we update the writing again throughout the paper to clarify all descriptions related to "scaling", ensuring this distinction is more explicit. Additionally, we will revise the title of the paper to replace "scaling" with "large" to further avoid any ambiguity, i.e., *Large Diffusion Language Models via Adaptation from Autoregressive Models* in the final version.
> > >
> > > Thank you again for pointing this out! We hope [this revised manuscript](https://openreview.net/pdf?id=j1tSLYKwg8) address your concerns.

---

### Official Review · Reviewer_XhLb · 2024-11-08

**Soundness:** 3
**Presentation:** 3
**Contribution:** 2
**Rating:** 6
**Confidence:** 4

**Summary:**

The paper studies diffusion language models (DLMs) and how they could be a good alternative to autoregressive (AR) LMs for generating text. It highlights the lack of fair comparisons on language modeling benchmarks and the challenges of training DLMs at scale. The authors propose adapting open-source AR models to build text diffusion models. By demonstrating the connections between AR and diffusion modeling objectives, they introduce a simple continual pre-training approach for training these models. The study shows that popular AR LMs can be converted into diffusion models (DiffuGPT and DiffuLLaMA) using less than 200B tokens for training, outperforming earlier DLMs and being competitive with their AR counterparts. The paper contributes a suite of DLMs capable of generating fluent text, performing in-context learning, text in-filling and following instructions.

**Strengths:**

The paper presents a simple and effective approach to adapting AR models to build DLMs, bridging the gap between the two modeling paradigms, while the proposed continual discrete diffusion pre-training approach for training diffusion models is practical and effective.
The proposed models, DiffuGPT and DiffuLLaMA, show competitive performance with AR models and outperform earlier DLMs in generating fluent text, perform in-context learning, filling in the middle without prompt re-ordering, and following instructions.
The writing, organization and presentation of the paper are also good; I enjoyed reading the manuscript.

**Weaknesses:**

Overall, I appreciate the achievement of well-performing (discrete) diffusion LMs at scale, but I do have some concerns regarding novelty and evaluation.

**About building DLMs by scaling and adapting from pre-trained large-scale LMs:**
Building (discrete) diffusion LMs from pre-trained Masked LMs has been studied in [1], while scaling and instruct-tuning (discrete) diffusion by adapting large-scale pre-trained LMs (mainly Masked-LMs while they also attempted to use AR-LMs/Llama1 to test reasoning) has also been studied in [2]. While this paper makes a solid step forward and effectively improves the performance of DLMs by leveraging AR LLMs, the authors should have thoroughly discussed the connections and major differences of their proposed approach with these prior works in the introduction or at least method/related work sections. Can the authors elorabrate the substantial differences and new contributions introduced by this paper?

**About progressively adapting AR text generation models into non-AR models:**
Two core techniques introduced in this paper are "Attention Mask Annealing" and "Shift Operation" to effectively reprogram an AR LM into a DLM. However, a similar approach has also been proposed in [3] in the case of non-autoregressive machine translation, which should have been discussed in this paper.

**Some claims are not supported by evaluations, hence the true advantages of DLMs are not convincing:**
The paper introduces the motivation of pushing the limits of DLMs by arguing that
> Notable challenges include difficulties in **future planning** (Bachmann & Nagarajan, 2024; Hu* et al., 2024; Xie et al., 2024) and **self-correction** (Huang et al., 2024). These constraints have spurred researchers to explore alternative architectures for next-generation LLMs.

and accordingly,
> Notably, DLMs exhibit promising capabilities in **intermediate token correction** (Ye et al., 2024) and **global planning** (Zhang et al., 2023), thereby addressing key limitations inherent in the AR approach.

However, there is no clear evidence in this paper, not even qualitative ones or case studies, to showcase that DiffuGPT and DiffuLLaMA possess such promising capabilities of self-correction and global planning. Only evaluating on language understanding benchmarks with weaker performance compared to AR-LLMs (Table 1) cannot convince the community why we need really DLMs as an alternative to AR LMs and as a new LLM paradigm (I think this is the most important question every DLM paper needs to address). Plus, the in-filling performance of AR-LMs was unfairly assessed as they are only provided with prefixes, according to Table 1's caption. Thus, the paper still fails to demonstrate the unique, distinct advantages of DLMs over AR-LMs. I strongly suggest the authors provide more evidence to differentiate DLMs' strengths in these regards, which can also support the motivations of this work in the introduction.

Overall, I think this manuscript stands a good contributions to DLM research, and I believe it could be greatly improved if authors can address the aforementioned concerns.

----

[1] DiffusionBERT: Improving generative masked language models with diffusion models. ACL 2023

[2]  Diffusion language models can perform many tasks with scaling and instruction-finetuning. Arxiv 2023

[3] Fine-Tuning by Curriculum Learning for Non-Autoregressive Neural Machine Translation. AAAI 2020

**Questions:**

See weaknesses.

---

> ### Author Response · Authors · 2024-11-22
> **Response to reviewer XhLb (1/2)**
>
> We sincerely thank Reviewer XhLb for acknowledging our contributions to the DLM research. We wish to address your concerns by providing detailed responses to each of your comments.
>
> > Weak1 & 2: The difference with previous work
>
> For the two works DiffusionBert [1] and scaling MLM [2], we discussed them in the related work (line 511). The substantial differences between [1][2] and our work include:
> 1. Both [1] and [2] primarily focus on mask language models (such as BERT and XLM), while **our work is unique in its successful adaptation of casual language models (AR)**. Compared with MLMs, resources allocated to AR large language models (LLMs) are significantly more today, and the scalability of LLM is more promising.
> 2. Despite of the benefits of adapting AR LLMs, it is non-trivial to achieve this. As discussed in [2], attempts to develop DLMs from LLaMA were unsuccessful, **indicating that adapting AR LLMs to DLMs is more complex than adapting MLMs to DLMs**. Theoretically, the gap between causal LLM and DLM is more pronounced and to address this, we bridge this gap in sec 3.2 and propose a series of techniques in sec 3.3. We are the first to successfully adapt AR LLM to a DLM.
> 3. The evaluations conducted in [1] and [2] do not compare to state-of-the art AR LLMs (they mostly focus on test set PPL and specific tasks like translation). Our work develops a comprehensive evaluation across diverse tasks including QA/multiple-choices/code/math, **facilitating a consistent assessment alongside popular LLMs**. We also evaluate in zero-shot, few-shot and fine-tuning settings.
>
> For paper [3], thanks for pointing this related work out. After reading this paper, we share similar ideas about leveraging existing AR models but the technical details differ: they mix the input sequence and the attention mask of AT/NAT, while we directly shift the input sequence and use the intermediate state between the two attention masks. Besides, diffusion models and NAT are easier to train on labeled data (like seq2seq tasks- machine translation). However, training DLMs on unlabeled data is more challenging due to the sample efficiency for discrete diffusion training [11].
>
> Overall, thank you for pointing out the need for thorough discussion and we have added more details in the introduction and related work. Please check our updated [manuscript](https://openreview.net/pdf?id=j1tSLYKwg8).

---

> ### Author Response · Authors · 2024-11-22
> **Response to reviewer XhLb (2/2)**
>
> > Weak3: True advantages of DLMs
>
> The advantage of DLMs has been discussed and fairly compared in the previous peer-reviewed work, like in math [4] and RL policy [5]. Additionally, there are concurrent studies under submission to ICLR, including [6] and [7]. These works highlight the unique strengths of diffusion models in search and planning tasks, largely due to their bi-directional modeling capabilities. In our work, we primarily focus on leveraging the pre-trained weights of large language models (LLMs) to efficiently train DLMs. To show its strength, we validate that DiffuLLaMA/DiffuGPT outperforms LLaMA/GPT2 on the GSM8K fine-tuning and infilling setting. To further examine their advantages mentioned in [4] (self-correction) and [6] (global planning), we follow [4] to conduct qualitative analysis and **observe similar self-correction ability in DLMs**. We show the last few steps of sampling trajectories and can observe that DLMs will refine the intermediate numbers without the restriction of left-to-right:
> ```
> <<3*15=45>> <<4*45=180>> <<180+300=00>> ####  00</s>
> <<3*15=45>> <<4*45=180>> <<180+400=000 #### #### 000</s>
> <<3*15=45>> <<4*45=180>> <<180+400=400>> #### 480</s>
> <<3*15=45>> <<4*45=180>> <<180+300=500>> #### 580</s>
> <<3*15=45>> <<4*45=180>> <<180+300=580>> #### 580</s>
> <<3*15=45>> <<4*45=180>> <<180+300=480>> #### 580</s>
> <<3*15=45>> <<4*45=180>> <<180+300=480>> #### 580</s>
> <<3*15=45>> <<4*45=180>> <<180+300=480>> #### 480</s>
> <<3*15=45>> <<4*45=180>> <<180+300=480>> #### 480</s>
> ```
> For global planning, [8] only shows the potential global planning capacity but not verify it and our model settings are different. Thus we follow [6] to finetune DLMs on counting down (CD) datasets. CD is a mathematical reasoning challenge and is a generalized version of the game of 24 which even advanced models such as GPT-4 struggle with [12]. We compare DiffuGPT with other AR baselines with different model size, **showing the global planning advantage of DLMs as demonstrated in [6]**.
> | Models           | CD4  |
> |------------------|------|
> | GPT2-scratch (85M)| 45.8 |
> | LLaMA FT (13B)   | 51.1 |
> | SoS (250M)       | 54.2 |
> | DiffuGPT-M       | 87.5 |
>
> While we acknowledge that the generalization capabilities of DLMs remain an open question, further exploration of their performance across a broader range of downstream tasks—particularly using 7B-scale DLMs—could be a promising direction for future work.
>
> For **infilling tasks**, we also try to query LLaMA model by prompting “given the <prefix> and <suffix>, please answer the <middle> part”, which includes both prefix and suffix information. However, this approach is no better than performing text completion with solely prefixes, likely because the LLaMA model needs tuning for filling in the middle (like codellama and [9]). Additionally, [9] notes that using autoregressive (AR) models for infilling presents challenges, such as prompting difficulties and repetition. In contrast, DLMs are naturally suited for this task, as they are trained to handle masked inputs [10], which is a key advantage.
>
> Additionally, we conduct a controlled experiment by training both AR and DLMs on 100M tokens from the Starcoder dataset, using CodeLLaMA as the base model and evaluating performance on HumanEval infilling. We finetune CodeLLaMA autoregressively with FIM [9] in both suffix-prefix-middle (SPM) and prefix-suffix-middle (PSM) formats. Our results in the following table show that DiffuCodeLLaMA outperforms PSM, suggesting that prompt format affects AR models but not DLMs. Furthermore, we believe that training on more than 100M tokens, which is relatively small, could enhance performance.
>
> |Models | Pass@1 HumanEval|
> |-----|------|
> |CodeLLAMA FT (FIM SPM) | 0.80 |
> |CodeLLAMA FT (FIM PSM) | 0.74 |
> |Diffu-CodeLLaMA (Ours) |0.76|
>
> **Reference**
>
> [1] DiffusionBERT: Improving generative masked language models with diffusion models. ACL 2023
>
> [2] Diffusion language models can perform many tasks with scaling and instruction-finetuning. Arxiv 2023
>
> [3] Fine-Tuning by Curriculum Learning for Non-Autoregressive Neural Machine Translation. AAAI 2020
>
> [4] Diffusion of Thoughts: Chain-of-Thought Reasoning in Diffusion Language Models. NeurIPS 2024
>
> [5] Diffusion Policy Visuomotor Policy Learning via Action Diffusion. RSS 2023
>
> [6] Beyond Autoregression: Discrete Diffusion for Complex Reasoning and Planning. https://openreview.net/forum?id=NRYgUzSPZz
>
> [7] Implicit Search via Discrete Diffusion: A Study on Chess. https://openreview.net/forum?id=A9y3LFX4ds
>
> [8] PLANNER: Generating diversified paragraph via latent language diffusion model. NeurIPS 2023
>
> [9] Efficient Training of Language Models to Fill in the Middle. OpenAI
>
> [10] Diffusion Models for Video Prediction and Infilling. TMLR 2022
>
> [11] Diffusion Language Models https://sander.ai/2023/01/09/diffusion-language.html
>
> [12] Stream of search (sos): Learning to search in language. COLM 2024

---

> > ### Comment · Reviewer_XhLb · 2024-11-25
> >
> > I truly appreciate the detailed response from the authors. I believe the new discussion and results are essential to the soundness of this work. I decided to increase my rating to 6.

---

### Author Response · Authors · 2024-11-22
**A general response to all reviewers**

Dear reviewers:

We deeply appreciate your insightful feedback and valuable suggestions. Based on your reviews, we have made thorough revisions to our [manuscript](https://openreview.net/pdf?id=j1tSLYKwg8).

**Contribution Summary**

We develop a method to adapt AR models into diffusion models and release pretrained DLMs. Our focus is on comparing DLMs to AR models on tasks where AR models excel with large-scale training on unlabeled data. In the future, these base DLMs can be used for instruction tuning, controlled text generation, and creating joint language-vision models.

**Our Strengths**

Thank you for recognizing our strengths, including:

1. We tackle an important, previously unexplored problem in the DLM community (XhLb, ZRNS, rZH1).
2. Our evaluation is solid and comprehensive (rZH1, 87p8).
3. Our DLMs demonstrate advantages over AR models (XhLb, ZRNS).

**Our Key Revisions**

1. We have added a thorough discussion on the connections and differences between our work and previous studies ([Line 55](https://openreview.net/pdf?id=j1tSLYKwg8), [Line 525](https://openreview.net/pdf?id=j1tSLYKwg8)).
2. We highlight the advantages of DLMs through (i) a new case study on self-correction, (ii) global planning on countdown data, and (iii) additional code infilling experiments ([Appendix C.3](https://openreview.net/pdf?id=j1tSLYKwg8)).
3. We clarified misunderstandings regarding our dataset selection process and experiment results analysis ([Line 370](https://openreview.net/pdf?id=j1tSLYKwg8), [Appendix B.1](https://openreview.net/pdf?id=j1tSLYKwg8)).
4. We included additional controllable experiment results, and compared our results with autoregressive LMs that are continue-pretrained on the same corpus ([Appendix C.4](https://openreview.net/pdf?id=j1tSLYKwg8)).
5. We added ELBO statistics for GSM tasks and DiffuLLaMA zero-shot results for different T ([Appendix C.2](https://openreview.net/pdf?id=j1tSLYKwg8), [C.5](https://openreview.net/pdf?id=j1tSLYKwg8)).

Thank you again for your contributions to improving our work. We are happy to address any further concerns or queries.

---

### Public Comment · ~Subham_Sekhar_Sahoo1 · 2025-04-26
**Request for Citation of Prior Work on Training Objective (Eq. 6)**

Thanks for the great work! I just wanted to kindly point out that the training objective described in `Eq. 6` was previously proposed in `Sahoo et al. (2024)`, along with related developments in Shi et al. (2024) and Ou et al. (2025). I would greatly appreciate it if the authors could acknowledge this prior work by citing this reference in their manuscript.

```
@inproceedings{
sahoo2024simple,
title={Simple and Effective Masked Diffusion Language Models},
author={Subham Sekhar Sahoo and Marianne Arriola and Aaron Gokaslan and Edgar Mariano Marroquin and Alexander M Rush and Yair Schiff and Justin T Chiu and Volodymyr Kuleshov},
booktitle={The Thirty-eighth Annual Conference on Neural Information Processing Systems},
year={2024},
url={https://openreview.net/forum?id=L4uaAR4ArM}
}
```

---

> ### Public Comment · ~Shansan_Gong1 · 2025-07-03
>
> Thank you for suggestions and sorry for the late reply! MDLM is a great work! This work is already added in our camera ready version!

---

### Meta-Review · Area_Chair_Ps2V · 2024-12-20

**Metareview:**

The authors study diffusion language models. In particular, they come up with compute efficient ways of adapting a pre-trained AR model into a diffusion one, and show that these resulting adapted diffusion models are performant and generally comparable to their original AR equivalents.

While adapting an AR model to diffusions is not amazingly novel, the reviewers generally agree that the results are compelling, and of interest to the diffusion language modeling community. The evaluations are also solid enough that these results can be used as foundations for future work, and its likely that these types of techniques will be used to more computationally efficiently study diffusion models.

**Additional Comments On Reviewer Discussion:**

Authors gave an extensive rebuttal, and resolved many of the reviewer comments e.g. for writing.

---

### Decision · Program_Chairs · 2025-01-22

Accept (Poster)